# Bearing Fault Feature Extraction Method Based on Adaptive Time-Varying Filtering Empirical Mode Decomposition and Singular Value Decomposition Denoising

**Xuezhuang E, Wenbo Wang *** and **Hao Yuan**

Hubei Province Key Laboratory of System Science in Metallurgical Process, Wuhan University of Science and Technology, Wuhan 430081, China; exuezhuang@wust.edu.cn (X.E.); yuanhao2022@wust.edu.cn (H.Y.)
* Correspondence: wangwenbo@wust.edu.cn

**Abstract:** Aiming to address the difficulty in extracting the early weak fault features of bearings under complex operating conditions, a fault diagnosis method is proposed based on the adaptive fusion of time-varying filtering empirical mode decomposition (TVF-EMD) modal components and singular value decomposition (SVD) noise reduction. First, the snake optimization (SO) technique is used to optimize the TVF-EMD algorithm in order to determine the optimal parameters that match the input signal. Then, the bearing signal is divided into a number of intrinsic mode functions (IMFs) using TVF-EMD in order to reduce the nonlinearity and non-stationary characteristics of the fault signal. An index for the envelope fault information energy ratio (EFIER) is created to overcome the drawback of there being too many IMF components after TVF-EMD decomposition. The IMF components are ranked in descending order according to the EFIER, and they are fused according to the maximum principle of the energy ratio of envelope fault information until the optimal fusion component is determined. Finally, the fault feature is extracted when the optimal fusion component is denoised using SVD. Two measured bearing fault signals and simulation signals are used to validate the performance of the proposed method. The experimental findings demonstrate that the approach has good sensitive feature screening, fusion, and noise reduction capabilities. The proposed method can more precisely extract the early fault features of bearings and accurately identify fault types.

**Keywords:** time-varying filtering empirical mode decomposition; snake optimization; fault diagnosis; rolling bearings

## 1. Introduction

Rolling bearings are the most crucial components of rotating machinery equipment and are extensively used in mechanical engineering because of their compact structure, ease of maintenance, high mechanical efficiency, long service life, and stable and reliable quality [1,2]. According to statistics, nearly one-third of the faults in rotating machinery using rolling bearings are caused by bearing faults [3,4]. Therefore, it is of great significance to avoid major life and property accidents by timely identifying the running state of mechanical equipment, whether there is a fault in the bearing, and whether the time and location of the fault can be accurately judged [5]. At present, deep learning and transfer learning methods have also been widely used in the diagnosis of bearing faults [6,7].

Owing to its effectiveness in decomposing nonlinear and unstable signals, empirical mode decomposition (EMD) has often been used in the detection of rolling bearing faults [8]. However, there are some inherent problems in traditional empirical mode decomposition,

such as noise sensitivity and mode mixing, which limit the further improvement of its decomposition performance [9]. Recently, numerous EMD optimization and improvement solutions have been put forth and used, such as EEMD [10], CEEMD [11], CEEMDAN [12], VMD [13], etc.

In 2017, Li et al. [14] proposed the time-varying filtering empirical mode decomposition (TVF-EMD) algorithm as a solution to the modal aliasing issue of EMD. TVF-EMD utilizes instantaneous amplitude and frequency data to adaptively determine the local cut-off frequency [15,16]. The filtering process is completed using a nonuniform B-spline to construct an approximate time-varying filter. This technique is capable of fully adaptively analyzing nonlinear and non-stationary signals and can successfully increase decomposition performance at low sample rates. It has a better anti-modal aliasing ability and can more effectively separate different vibration characteristic signals than the EMD, EEMD, CEEMD, and VMD algorithms. Zhang et al. [17] presented a parameter-adaptive TVF-EMD technique based on the gray wolf optimization (GWO) algorithm. The goal function of the approach is the weighted kurtosis index, which is used to calculate the TVF-EMD bandwidth threshold and B-spline order. Ye et al. [15] created a novel composite measure called the sparse influence measure index (SIMI) as a parameter selection criterion and used the Bottle Sea Sheath Swarm Algorithm (SSA) to optimize the parameter combinations in the traditional TVF-EMD. Vashishtha et al. [12] suggested a deep learning-based strategy for identifying granular wheel flaws. First, amended gray wolf optimization (AGWO) is used to optimize the vibration signal based on the TVF-EMD filtering parameters, with the Kernel Estimation of Mutual Information (KEMI) serving as the objective function. The well-known IMF is used to prepare the dataset and build a scalogram. Convolutional neural network (CNN) training is used to categorize and identify the IMFs. Xu et al. [18] proposed a bearing failure detection technique based on TVF-EMD and a high-order energy operator. In this method, a single IMF component is selected using kurtosis for a fault analysis. Nonetheless, the selection of a single component can result in the loss of fault information due to the substantial number of components following TVF-EMD decomposition. Additionally, because the kurtosis index is susceptible to noise, the selection of IMF components can also easily be disrupted by noise. By integrating TVF-EMD and multipoint optimal minimal entropy deconvolution adjusted (MOMEDA), Xu et al. [19] proposed a bearing defect feature identification method. This method selects the optimal IMF component from the TVF-EMD decomposition findings using kurtosis, then improves the MOMEDA fault feature of the optimal IMF component, and finally analyzes its envelope spectrum characteristics. However, this method also has shortcomings; for example, the kurtosis index is sensitive to noise, and a single IMF component easily loses fault information.

Therefore, an appropriate intelligent optimization technique must be used to optimize the TVF-EMD algorithm's parameters in order to extract fault features effectively. The ideal solution is directly impacted by the objective function choice, so it is imperative to design an effective measurement index for identifying weak fault features amidst noise interference. Based on this, devising a suitable modal component fusion method to integrate the components of TVF-EMD is also essential. This approach aims to overcome the shortcomings of the overly dispersed and insufficiently focused fault feature information following TVF-EMD decomposition, thereby optimizing the TVF-EMD fault detection method. Therefore, this study designs a new sensitive component selection index—the envelope fault information energy ratio (EFIER)—to screen IMF components. After TVF-EMD decomposition, the IMF components are fused using the EFIFR maximum principle for obtaining the optimal OC component. To additionally mitigate the impact of noise, singular value decomposition (SVD) is employed to denoise the optimal OC component,

and, finally, fault feature extraction is performed on the noise-reduced signals. The main contributions of the proposed method are as follows:

(1) Aiming at the difficulty of choosing the parameter combination of TVF-EMD algorithm, the snake optimizer (SO) algorithm [20] was used to optimize the bandwidth threshold $\xi$ and B-spline order $n$ of TVF-END to obtain the best decomposition performance.

(2) To solve the problem of the loss of effective fault characteristic information due to excessive modes after the decomposition of TVF-EMD, a new index of the envelope fault information energy ratio (EFIER) is proposed.

(3) The IMF components were arranged in descending order by the EFIER, and the IMF components were fused successively according to the increasing principle of the EFIER to construct the optimal fusion component.

(4) In order to further reduce the noise interference in the optimal fusion component, an improved SVD is used to reduce the noise of the optimal fusion component, which effectively enhances the extraction ability of early fault features.

The remainder of this paper is structured as follows: Section 2 provides a succinct overview of SO, TVF-EMD, and SVD theories. Detailed introductions to the methods proposed in this paper are presented in Section 3. Section 4 confirms the method's efficacy by utilizing simulation signals, and Section 5 confirms its superiority through comparative experiments and verification using two bearing signals. Section 6 draws the conclusions.

## 2. Theoretical Basis

### 2.1. SO

The snake optimizer (SO) [20] is a novel meta-heuristic algorithm that mimics the unique mating behavior of snakes, which was proposed in 2022. The SO algorithm and mathematical model are presented below.

(1) Initialization

Initially, a uniformly distributed random population is produced via the SO method and initialized with the following mathematical description:

$$X_i = X_{\min} + r * (X_{\max} - X_{\min}) \tag{1}$$

where $X_i$ represents the ith snake's position; $r$ denotes a random number in the interval $[0, 1]$; and $X_{\max}$ and $X_{\min}$ represent the problem's upper and lower bounds, respectively.

(2) Dividing the population into female and male groups

The population is split into two groups, one for each gender, with a ratio of 1:1. $X_i$ can now be split into two halves, $X_i^m$ and $X_i^f$, where $m$ and $f$ represent male and female, respectively. The temperature $Temp$ and the quantity of food $Q$, which are determined using the following equations, respectively, have a major impact on the exploration and exploitation phases of the SO:

$$Temp = \exp(-t/T) \tag{2}$$

$$Q = c_1 * \exp((t - T)/T) \tag{3}$$

where $t$ represents the current iteration number; $T$ represents the maximum iteration number; and $c_1 = 0.5$.

(3) Search phase (no food)—global search

When $Q < 0.25$, the SO enters the exploration phase, where it chooses a random place and updates its position in the search for food. The random exploration formula is as follows:

$$X_i^m = X_{rand}^m(t) \pm c_2 * A_m * ((X_{\max} - X_{\min}) * rand + X_{\min}) \tag{4}$$

where $X_i^m$ is the location of the male snake; $X_{rand}^m$ is the location of the randomly selected male snake; $rand \in [0,1]$; $c_2 = 0.05$; and $A_m$ is the ability of the male to find food, which is calculated as follows:

$$A_m = \exp(-f_{rand}^m / f_i^m) \tag{5}$$

where $f_{rand}^m$ is the fitness value of $X_{rand}^m$ and $f_i^m$ is the fitness value of $X_i^m$ (4) Development phase (with food)

In contrast to the exploration mode, the development phase of the SO is separated into three modes: proximity to prey (food), fighting, and mating. Each of these modes is influenced by temperature T, as well as the quantity of food Q.

The SO looks for food when $Q > 0.25\,\&\,Temp > 0.6$, and the formula for updating its position is as follows:

$$X_{i,j}(t+1) = X_{food} \pm c_3 * Temp * rand * (X_{food} - X_{i,j}(t)) \tag{6}$$

where $X_{i,j}$ is the entire population (males and females); $X_{food}$ is the global optimal position; and $c_3 = 2$. When $Q > 0.25\,\&\,Temp > 0.6$, the snake is in fighting mode or mating mode [20].

*2.2. TVF-EMD*

2.2.1. Basic Theory of TVF-EMD

In the TVF-EMD method, the low-pass filter constructs the cutoff frequency with time to complete the iterative mean removal operation existing in other modal decomposition methods, and the local narrowband signal replaces the IMF as the iterative stop condition. TVFEMD is mainly completed through the following three steps [14].

(1) Calculate the local cutoff frequency

Firstly, the instantaneous amplitude $A(t)$ and the instantaneous frequency $\varphi'(t)$ of the signal $x(t)$ are obtained by the Hilbert transform, and then the difference between their local maximum and local minimum values is obtained to obtain $\eta_1(t)$ and $\eta_2(t)$. At the same time, the instantaneous mean $a_1(t)$ and the instantaneous envelope $a_2(t)$ are obtained by filtering the input signal with a time-varying filter. Then, the local cutoff frequency $\varphi'_{bia}(t)$ is calculated.

$$\varphi'_{bia} = \frac{\eta_2(t) - \eta_1(t)}{4a_1(t)a_2(t)} \tag{7}$$

(2) The signal is reconstructed as $h(t)$ according to the adjusted intercept frequency. Let

$$h(t) = \cos\left[\int \varphi'_{bia}(t)dt\right] \tag{8}$$

(3) Determine whether the stop criteria is met. Calculate the stop criterion.

$$\theta(t) = \frac{B_{Loughlin}}{\varphi_{avg}(t)} \tag{9}$$

$$\varphi_{avg}(t) = \frac{a_1^2(t)\varphi'_1(t) + a_2^2(t)\varphi'_2(t)}{a_1^2(t) + a_2^2(t)} \tag{10}$$

$$\varphi'_1(t) = \frac{\eta_1(t)}{2a_1^2(t) - 2a_1(t)a_2(t)} + \frac{\eta_2(t)}{2a_1^2(t) + 2a_1(t)a_2(t)} \tag{11}$$

$$\varphi'_2(t) = \frac{\eta_1(t)}{2a_2^2(t) - 2a_1(t)a_2(t)} + \frac{\eta_2(t)}{2a_2^2(t) + 2a_1(t)a_2(t)} \tag{12}$$

$$B_{Loughlin}(t) = \sqrt{\frac{\alpha\prime_1^2(t) + \alpha\prime_2^2(t)}{\alpha_1^2(t) + \alpha_2^2(t)} + \frac{\alpha_1^2(t)\alpha_2^2(t)(\varphi\prime_1(t) - \varphi\prime_2(t))^2}{(\alpha_1^2(t) + \alpha_2^2(t))^2}} \tag{13}$$

Based on Equations (9)–(13), $\theta(t)$ is calculated. The remaining signal $x(t)$ can be categorized as a narrow-band signal if $\theta(t)$ is greater than or equal to the specified bandwidth threshold $\xi$; otherwise, steps 1 to 8 are repeated while setting $x(t) = x(t) - s(t)$.

### 2.2.2. Mode Separation Ability of TVF-EMD

We designed a multi-component amplitude modulation frequency modulation signal to simulate the fault signal, and its expression is as follows:

$$\begin{cases} x(t) = x_1(t) + x_2(t) + x_3(t) \\ x_1(t) = (0.3 + 0.3\sin(20\pi t)) \times \cos(600\pi t + 1.5\sin(20\pi t))) \\ x_2(t) = (1 + \sin(20\pi t)) \times \cos(300\pi t + 0.5\sin(10\pi t)) \\ x_3(t) = (0.5 + 0.5\sin(10\pi t)) \times \cos(100\pi t + 0.8\sin(10\pi t)) \end{cases} \tag{14}$$

The dominant frequencies of each component are 50 Hz, 150 Hz, and 300 Hz, respectively, and the sampling frequency is 1000 Hz. The time domain waveform and frequency domain waveform of the synthesized simulation signal are shown in Figure 1.

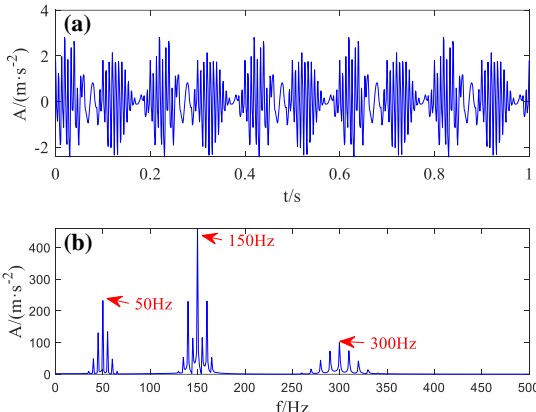

**Figure 1.** Simulated signal: (**a**) time domain waveform; (**b**) frequency spectrum.

TVF-EMD was used to decompose the signal. The bandwidth threshold $\xi$ of TVFEMD was set to 0.1, and the B-spline order $n$ was set to 16. The decomposition results were shown in Figure 2. It can be seen that TVFEMD effectively separates the three frequency components of the simulation signal. The results show that TVFEMD can decompose the multi-component FM analog fault signal into several modes effectively, reduce the mode chaos phenomenon, and reduce the non-stationarity of the signal.

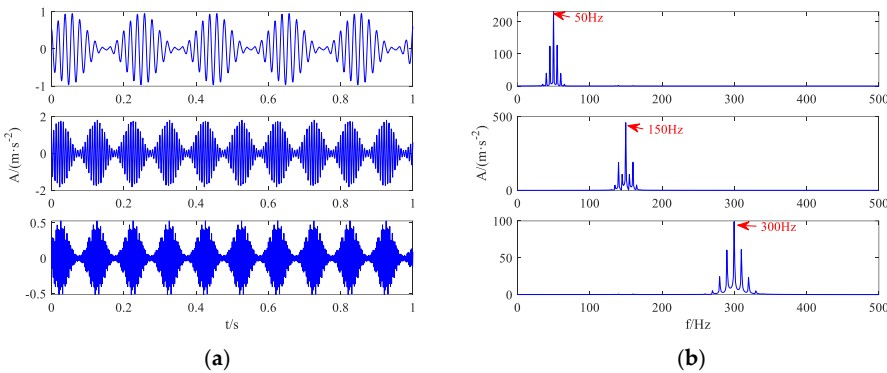

**Figure 2.** Decomposing results of TVF-EMD: (**a**) three IMFs; (**b**) spectrograms of IMFs.

### 2.3. Improved SVD Noise Reduction

Singular value decomposition is a matrix decomposition approach for time series signals. Suppose the noise-contaminated signal is

$$y(i) = x(i) + n(i), i = 1, 2, \ldots, N \tag{15}$$

where $x(i)$ represents the initial signal and $n(i)$ denotes noise. Then, the noise contaminated signal's Hankel matrix is as follows [21]:

$$H = \begin{bmatrix} y(1) & y(2) & \cdots & y(K) \\ y(2) & y(3) & \cdots & y(K+1) \\ \vdots & \vdots & \cdots & \vdots \\ y(L) & y(L+1) & \cdots & y(N) \end{bmatrix} \tag{16}$$

where $K = N - L + 1$ and $L = \lfloor N/3 \rfloor$. According to SVD theory, the decomposition result of matrix $H$ is as follows:

$$H = UDV^T = \sum_{i=1}^{r} \sigma_i u_i v_i^T \tag{17}$$

$U$ and $V$ denote orthogonal matrix of $L \times L$, and $r = \min(K, L)$. The singular value satisfies $\lambda_1 > \lambda_2 > \cdots > \lambda_r > 0$, and $u_i$ and $v_i$ represent $K$-dimension and $L$-dimension column vectors.

According to SVD theory, the first $p$ larger singular values reflect the effective signal, while the last $r - p$ smaller singular values reflect the noise component. Based on this, the $r - p$ smaller singular values can be set to 0, and only the first $p$ singular values can be retained. After SVD reconstruction, the denoised bearing vibration signal can be obtained.

The effectiveness of SVD denoising depends on the selection of the effective rank order of the singular values. In the research process of this manuscript, efforts were made to preserve fault information as much as possible while removing noise. We used the curvature spectrum method to select the effective rank order of the singular values. Finding the turning point of the singular value curve is the key to distinguishing between noise and vibration information. In order to automatically select suitable turning points on the singular value curve, we use the curvature spectrum to describe the changes in singular values, thereby finding the range of singular values accurately when reconstructing fault signal components.

The formula for calculating the curvature spectrum $C$ of the singular value curve is as follows:

$$C_i = \frac{|y''(i)|}{\left(1 + y(i)^2\right)^{3/2}} \tag{18}$$

$$y''(i) = \lambda_{i+1} - 2\lambda_i + \lambda_{i-1} \tag{19}$$

where $i = 2, 3, \cdots, p-1$; $y'(i)$ denotes first derivative; and $y''(i)$ denotes the second derivative. Let $C_1 = 0$, and sequence $\{C_1, C_2, \cdots, C_{p-1}\}$ is chosen as the curvature spectrum of the singular values.

This reflects the turning point of the singular value curve. The larger the value of $C_i$, the greater the degree of transition of the singular value sequence at that point. The position k of the maximum peak $C_k$ of the singular value curvature spectrum can serve as the boundary point between the singular values of the fault signal and noise. If a = b, the formula for reconstructing the denoised fault signal using the first r singular values is as follows:

$$H = U_{L \times r} D_{r \times r} V_{K \times r}{}^T \tag{20}$$

## 3. Bearing Fault Diagnosis Method Based on Adaptive TVF-EMD and SVD

### 3.1. Envelope Fault Information Energy Ratio

To guarantee that the components containing more fault information can be effectively selected from the IMF components decomposed using TVF-EMD, this study proposes a new sensitive component selection index—the envelope fault information energy ratio (EFIER). The EFIER evaluates how much the envelope fault information energy contributes to the total envelope energy. When the impact component contained in the original component information is more abundant, the EFIER value is larger, and vice versa. The EFIER index can be calculated using the following formula:

$$EFIER = \frac{E(x)}{En(x)} \tag{21}$$

$$En(x) = \int_{-\infty}^{\infty} |x(t)|^2 dt \tag{22}$$

$$E(x) = \sum_{j=1}^{f_s/f} En(x(j * f * \frac{N}{f_s})) = \sum_{j=1}^{f_s/f} (x(j * f * \frac{N}{f_s}))^2 \tag{23}$$

where $En(x)$ represents the overall energy of the envelope signal; $E(x)$ denotes the envelope fault information energy; $N$ represents the number of sampling points; $f$ de notes the fault characteristic frequency; and $f_s$ denotes the sampling frequency. The EFIER measures the fault shock portion of the IMF component signal using the fault eigenfrequency. In the EFIER, the envelope energy of the impact component corresponding to a frequency that represents a fault characteristic is first determined, and the effective IMF component is then selected based on the proportion of the fault envelope energy to the total envelope energy. As the theoretical values of the fault eigenfrequency differ from the actual values, the energy of the peak close to the fault eigenfrequency is selected to serve as the envelope fault information energy. Calculating the index values of all conceivable fault types is essential if this index is to be utilized to monitor operating conditions or diagnose unidentified faults. It is possible to determine the characteristic frequencies of various fault types using Equations (21)–(23). The EFIER for different faults is calculated in turn when the fault characteristic frequency is calculated.

As an effective sensitive component selection index, the EFIER is robust to different noise intensities and burst pulses, in addition to being sensitive to fault feature information. The performance of the EFIER is compared with that of the kurtosis index, approximate entropy (ApEn) [22], and fuzzy entropy (FEn) [23] by simulated fault signals. The simulation signal of the outer ring fault is

$$\begin{cases} x(t) = x_0 e^{-2\pi\varepsilon f_n t} sin\left(2\pi f_n \sqrt{1-\varepsilon^2 t}\right) \\ y(t) = x(t) + n(t) \end{cases} \tag{24}$$

where the sampling frequency $f_s$ = 12 kHz, shift constant $x_0$ = 1, damping coefficient $\varepsilon$ = 0.1, bearing natural frequency $f_n$ = 2 kHz, samples number $N$ = 4096, characteristic frequency of the outer ring fault is $f_o$ = 100 Hz, and $n(t)$ is Gaussian white noise.

Gaussian white noise with SNR = −1 dB, −3 dB, −5 dB is added into the outer ring fault signal, respectively. The signals with different noise intensity are shown in Figure 3a–c. The EFIER, Kurt, ApEn, and FEn are calculated under different noise intensities, and the results are shown in Table 1. The EFIER values are 0.1396, 0.1394, and 0.1311, which remain relatively stable under different noise intensities. This indicates that the EFIER index has good robustness to weak noise. The Kurt values are 3.7199, 3.1888, and 2.9712, the ApEn values are 1.9532, 1.7350, and 1.5017, and the FEn values are 1.6868, 1.7999, and 1.9212,

respectively. The large fluctuations in Kurt, ApEn, and FEn indicate that their indexes are more susceptible to noise interference.

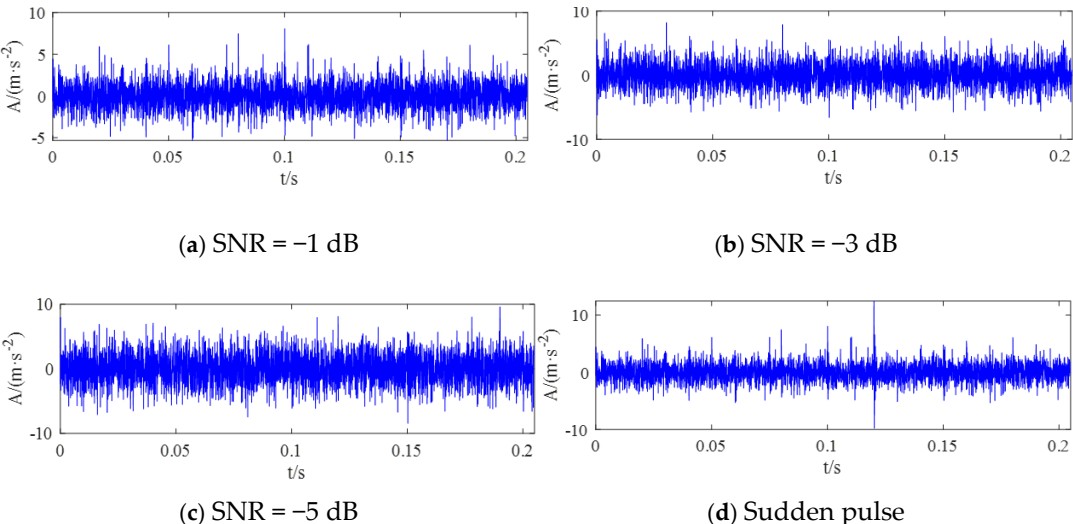

(**a**) SNR = −1 dB　　　　　　　　　　　　　　　　(**b**) SNR = −3 dB

(**c**) SNR = −5 dB　　　　　　　　　　　　　　　　(**d**) Sudden pulse

**Figure 3.** Fault signal with different noise intensities and sudden pulses.

**Table 1.** Comparison of evaluation indicators under different noise interferences.

| Index | Kurt | ApEn | FEN | EFIER |
|---|---|---|---|---|
| −1 dB | 3.7199 | 1.9532 | 1.6868 | 0.1396 |
| −3 dB | 3.1888 | 1.7350 | 1.7999 | 0.1394 |
| −5 dB | 2.9712 | 1.5017 | 1.9212 | 0.1311 |
| −5 dB + pulse | 5.2940 | 1.9472 | 1.6750 | 0.1387 |

Figure 3d shows the time-domain waveform generated by adding a pulse of amplitude $15\,\mathrm{m/s^2}$ based on Figure 3c. It can be seen from Table 1 that, after adding the pulse, the EFIER value is 0.1387 and the ApEn, Fen, and Kurt values are 1.9472, 1.6750 and 5.2940, respectively. It can be seen that the EFIER, ApEn, and FEn values under the influence of random pulses remain relatively stable, indicating that the EFIER, ApEn, and FEn indexes have sufficient robustness to random pulses. However, the fluctuation of the Kurt value is too large, indicating that the Kurt indicator is very sensitive to random pulses. When the vibration signal contains only random pulses without faults, it is often misjudged as containing rich fault information. From the above experimental results, it can be seen that the Kurt index is easily affected by random pulses and noise. Although the ApEn and FEn indexes have good robustness to random pulses, they are susceptible to noise interference. Moreover, the EFIER index has sufficient robustness to changes in environmental interference and has obvious superiority in determining whether vibration signals contain fault information.

### 3.2. Adaptive TVF-EMD Algorithm

The adaptive TVF-EMD algorithm's main concept is the iterative, repeated search of parameter combinations using an optimization algorithm to minimize the objective function and achieve parameter combination optimization. The objective function is

$$\begin{cases} fitness = \min_{\gamma=(\xi,n)} \{EFIER_i\} \\ s.t.\xi \in (0,1) \\ n \in [5,30], n \in Z \end{cases} \tag{25}$$

The individual steps of the adaptive TVF-EMD algorithm are as follows:

Step 1: Input the vibration signal x(t) and initialize the TVF-EMD and SO algorithm parameters (population size $N = 30$ and iteration number $L = 10$).

Step 2: Apply the updated parameter combination to TVF-EMD decomposition.

Step 3: Up until the maximum number of iterations L is achieved, compute and save the optimal fitness values and matching parameter combinations acquired in each iteration.

Step 4: Utilizing the globally optimal parameter combinations, perform TVF-EMD decomposition on the vibration signal.

### 3.3. IMF Component Fusion Based on EFIER

The TVF-EMD approach has too many IMF components after decomposition; hence, this study proposes a component fusion method based on the EFIER value to prevent the loss of effective fault feature information. The precise steps are as follows:

Step 1: All IMF components' EFIER values should be calculated, the highest value of the EFIER should be noted as MAX, and the matching IMF component should be designated as the OC component. Then, the IMF components should be arranged in descending order of the EFIER value.

Step 2: The OC component is merged with the first IMF component on the right to create the MF component, and the EFIER value of the MF component is determined in accordance with the IMF component's decreasing order arrangement.

Step 3: If the EFIER value of the MF component is less than the MAX, then the fusion is invalid and the fusion is stopped. If the EFIER value of the MF component is greater than the MAX, then the fusion is effective, the MAX and OC components are updated, and the IMF component continues to be fused to the right.

Step 4: By analogy, when the EFIER value of the fused MF component becomes lower than the MAX, the fusion is stopped and the optimal fusion component is the OC component that corresponds to the MAX.

A flowchart of TVF-EMD modal component fusion based on the EFIER maximum principle is shown in Figure 4.

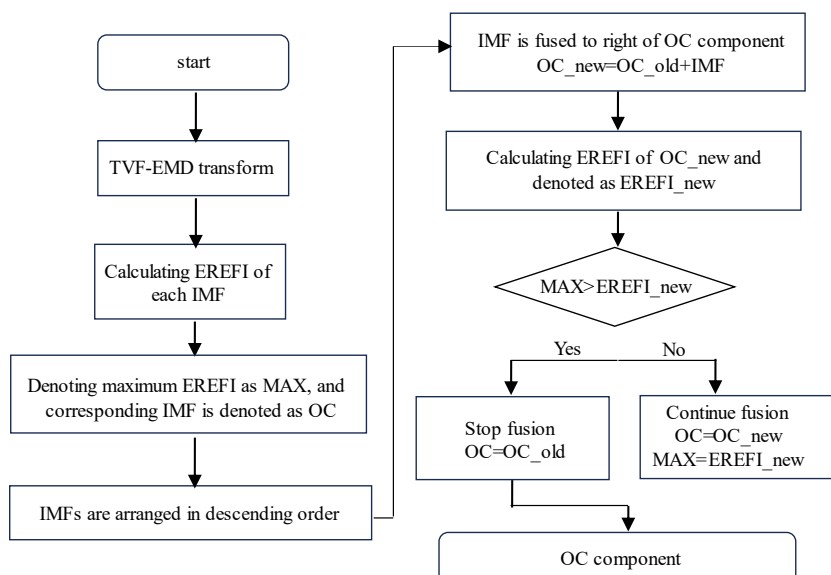

**Figure 4.** TVF-EMD modal component fusion based on EFIER maximum principle.

### 3.4. Bearing Fault Diagnosis Based on Adaptive TVF-EMD and SVD Denoising

After the TVF-EMD component is fused using the maximum principle of the envelope fault information energy ratio (EFIER), although the obtained optimal fusion component

contains a lot of fault information, the fusion signal still contains more noise at this time. In this study, the fusion signal is denoised using SVD, and the bearing fault features are subsequently extracted from the denoised signal using an envelope spectrum analysis. A flowchart of this method for bearing weak defect diagnosis is shown in Figure 5. The precise steps are as follows:

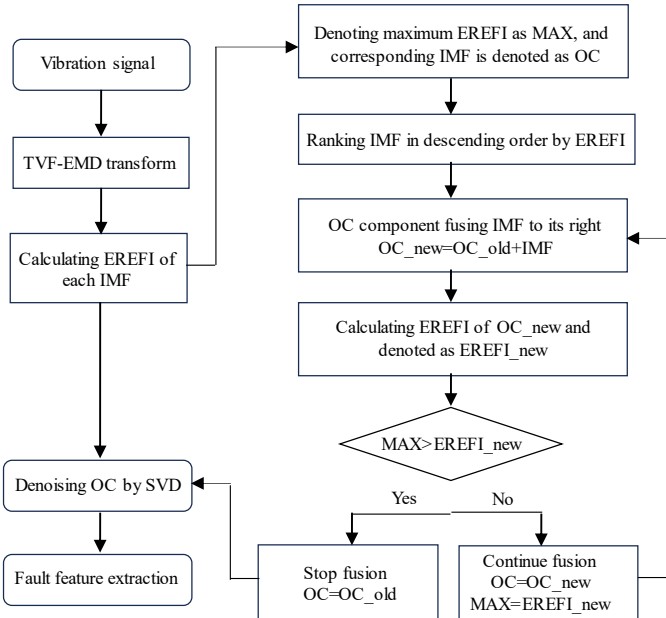

**Figure 5.** The calculation process of improved TVF-EMD and SVD methods.

Step 1: Set the TVF-EMD and SO algorithms' initial parameters.

Step 2: To determine the ideal parameter combination, optimize the TVF-EMD method using the SO algorithm.

Step 3: To break down the early defect vibration signal and obtain a number of component IMFs, substitute the ideal parameter combination $(\xi, n)$ into TVF-EMD.

Step 4: Determine each IMF component's EFIER value, and then arrange the components in descending order of this value.

Step 5: The IMF component is fused in accordance with the EFIER maximal principle to produce the ideal fusion component OC.

Step 6: Perform SVD denoising on the OC component.

Step 7: Analyze the noise-reduced components with envelope spectra.

## 4. Simulated Bearing Fault Experiment

We analyzed the effectiveness of the proposed SO-TVFEMD-EFIER-SVD method according to simulating bearing fault signals, and simulated bearing fault signals are generated by Equation (24).

### 4.1. Performance Analysis of SO-TVFEMD

A bearing outer ring fault simulating signal is generated by formula (24), where the sampling frequency $f_s$ = 12 kHz, shift constant $x_0$ = 1, damping coefficient $\varepsilon$ = 0.1, bearing natural frequency $f_n$ = 2 kHz, samples number $N$ = 4096, characteristic frequency of the outer ring fault is $f_o$ = 199.32; and $n(t)$ is Gaussian white noise with SNR = −1 dB. Comparing the SO-TVFEMD with the Whale Optimization algorithm [24], TVFEMD (WOA-TVFEMD), and the sparrow search algorithm [25], TVFEMD (SSA-TVFEMD), is achieved through the simulated bearing outer ring fault signal.

The time-domain waveform and envelope spectrum of the simulated signal are shown in Figure 6. It can be seen that when the periodic impact features in the time-domain waveform are completely submerged, it is unable to extract the impact features. The fault characteristic frequency components cannot be found in the envelope spectrum of the original signal and effective diagnosis cannot be carried out, so further analysis of the fault signal is needed. Using the maximized EFIER as the objective function, the SO, BWO, and SSA algorithms are employed to optimize the TVFEMD parameters, and the optimal parameter combination $[\xi, n]$ of TVFEMD is obtained. The population size of all algorithms is set to 30, and the maximum number of iterations is set to 500.

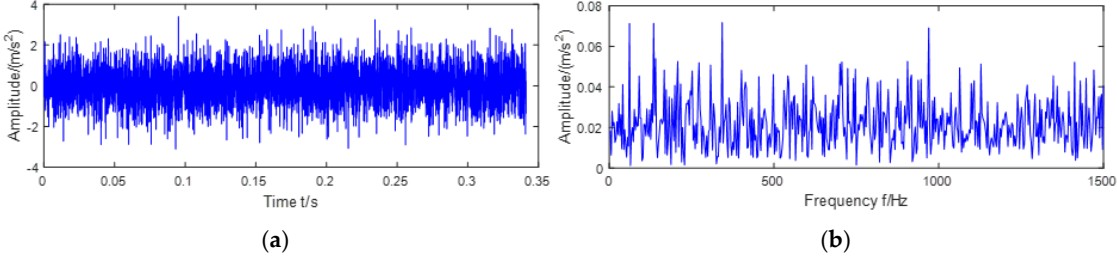

| (**a**) | (**b**) |

**Figure 6.** Outer ring fault simulation signal: (**a**) waveform; (**b**) envelope spectrum.

To eliminate the interference of randomness, we compared the mean convergence curves of 50 experiments using three methods. In order to further analyze the effect of the SO method on Gaussian white noise of different intensities, Gaussian white noise of $-3$ dB and $-5$ dB was added into the fault signal, respectively. For the Gaussian white noise of $-1$ dB, $-3$ dB, and $-5$ dB, the optimal fitness values obtained from 50 runs of three optimization algorithms are recorded, respectively. The average, optimal, worst, variance, and standard deviation of the running results of each algorithm are calculated, as shown in Table 2.

**Table 2.** Statistical index values of three algorithms (EFIER value).

| Algorithm | Max (%) | Min (%) | Range (%) | Mean (%) | Var | Std |
|---|---|---|---|---|---|---|
| SO ($-1$ dB) | 2.9874 | 2.9750 | 0.0125 | 2.9807 | $1.1560 \times 10^{-5}$ | 0.0034 |
| BWO ($-1$ dB) | 2.9462 | 2.9304 | 0.0158 | 2.9389 | $2.2090 \times 10^{-5}$ | 0.0047 |
| SSA ($-1$ dB) | 2.925 | 2.9078 | 0.0187 | 2.9168 | $3.4810 \times 10^{-5}$ | 0.0059 |
| SO ($-3$ dB) | 2.9761 | 2.9626 | 0.0135 | 2.9691 | $1.4440 \times 10^{-5}$ | 0.0038 |
| BWO ($-3$ dB) | 2.9300 | 2.9121 | 0.0179 | 2.9205 | $2.8090 \times 10^{-5}$ | 0.0053 |
| SSA ($-3$ dB) | 2.9096 | 2.8898 | 0.0198 | 2.9001 | $3.8440 \times 10^{-5}$ | 0.0062 |
| SO ($-5$ dB) | 2.9746 | 2.9611 | 0.0136 | 2.9679 | $1.5210 \times 10^{-5}$ | 0.0039 |
| BWO ($-5$ dB) | 2.9283 | 2.9112 | 0.0171 | 2.9203 | $2.6010 \times 10^{-5}$ | 0.0051 |
| SSA ($-5$ dB) | 2.9082 | 2.8885 | 0.0196 | 2.8978 | $4.2250 \times 10^{-5}$ | 0.0065 |

From the mean value of the EFIER in Table 2, it can be seen that the SO method has a higher convergence mean of the EFIER compared with the BWO and SSA methods. This indicates that for different intensities of Gaussian white noise ($-1$ dB, $-3$ dB, $-5$ dB), the solution accuracy of the SO method is more ideal and has strong search optimization ability. The standard deviation of the optimization results obtained by the SO algorithm is also the smallest, indicating that the SO algorithm has less fluctuation in the optimization results and better stationarity. So, the performance of the SO algorithm is significantly better than that of the WOA algorithm and SSA algorithm.

Based on the experimental data, the average single iteration time, average convergence times, and average convergence time for each algorithm are calculated. The results are shown in Table 3.

**Table 3.** The iteration time of the three algorithms.

| Algorithm | Single Iteration Time/s | Average Iterations | Average Convergence Time/s |
|---|---|---|---|
| SO ($-1$ dB) | 2.4467 | 11 | 26.9137 |
| BWO ($-1$ dB) | 2.7594 | 14 | 38.6316 |
| SSA ($-1$ dB) | 2.9496 | 15 | 47.1936 |
| SO ($-3$ dB) | 2.8107 | 13 | 36.5391 |
| BWO ($-3$ dB) | 3.1249 | 15 | 46.8735 |
| SSA ($-3$ dB) | 3.3247 | 17 | 56.5199 |
| SO ($-5$ dB) | 3.2418 | 17 | 55.1106 |
| BWO ($-5$ dB) | 3.6766 | 18 | 66.1788 |
| SSA ($-5$ dB) | 3.8194 | 20 | 76.3880 |

Compared with the WOA algorithm and SSA algorithm, it can be seen from the single iteration time that the SO algorithm has a smaller computational cost and a shorter average time per iteration. As the intensity of white noise increases, the single iteration time of the three optimization algorithms also increases accordingly, but the average single iteration time of the SO algorithm is still the smallest. Comparing the average convergence sum time, the SO algorithm has a faster convergence speed, so, compared with the WOA and SSA algorithms, its average convergence sum time is also the lowest. From the results of Tables 2 and 3, it can be seen that the SO method used in this paper has a smaller time cost and better convergence effect.

### 4.2. Performance Analysis of SO-TVFEMD-EFIER

To further validate the effectiveness of the proposed EFIER, we compared it with the commonly used fuzzy entropy (FEN) and kurtosis (Kurt) index. The simulated outer ring fault signal of Section 4.1 had Gaussian noise with SNR = $-5$ dB added. The methods of SO-TVFEMD-EREFI, SO-TVFEMD-FEn, and SO-TVFEMD-Kurt were used for their decomposition, and according to the EFIER, FEN and Kurt select the optimal IMF component and calculate the envelope spectrum of the optimal IMF component. The results of the three indexes are shown in Figure 7a–c.

Figure 7a shows the optimal IMF component and its envelope spectrum selected from the SO-TVFEMD-EFIER decomposition results. From the envelope spectrum of the selected optimal IMF component, it can be observed that the fault characteristic frequencies $f_o \sim 3f_o$ are very prominent, indicating that the EFIER is effective as a selection criterion and that the fault information separation effect is relatively excellent. Figure 7b shows the optimal component and its envelope spectrum selected from the SO-TVFEMD-FEN decomposition results. The fault characteristic frequency $f_o$, $2f_o$, $6f_o$ can be observed from the envelope spectrum of the optimal IMF component. However, a large amount of noise makes the fault characteristic frequency not clear enough. Figure 7c shows the optimal IMF component and its envelope spectrum after decomposition selected Kurt as the standard. The fault characteristic frequency $f_o$, $2f_o$ can be identified from the envelope spectrum. In the same way, due to the interference of a large amount of noise, the fault characteristic frequency is severely submerged and the fault characteristic frequency of the outer ring is not clear enough.

In summary, when using the EFIER indicator as a selection criterion, more IMF components that contain the most fault information can be selected accurately. Therefore, compared to the approximate entropy and kurtosis, the EFIER proposed in this paper has better performance as a TVFEMD decomposition and optimal IMF component selection index.

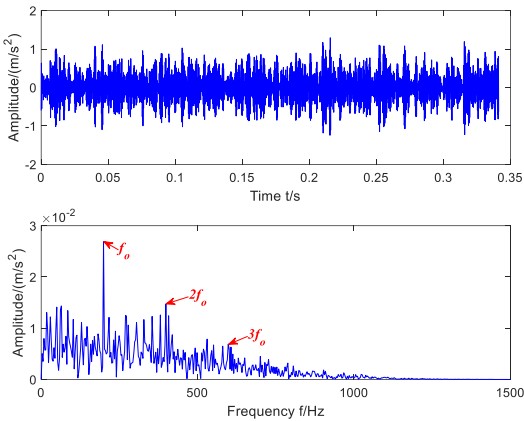

(**a**) Optimal component (above) and envelope spectrum (below) by EFIER index.

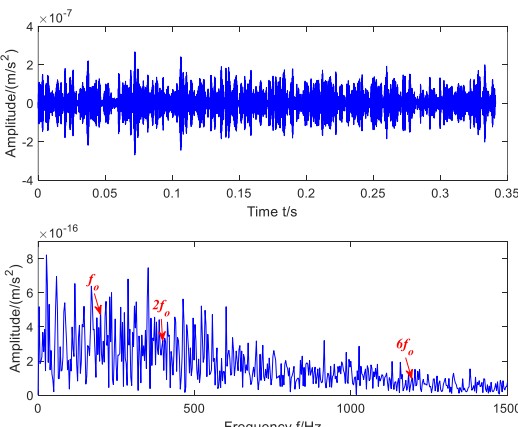

(**b**) Optimal component (above) and envelope spectrum (below) by FEN index.

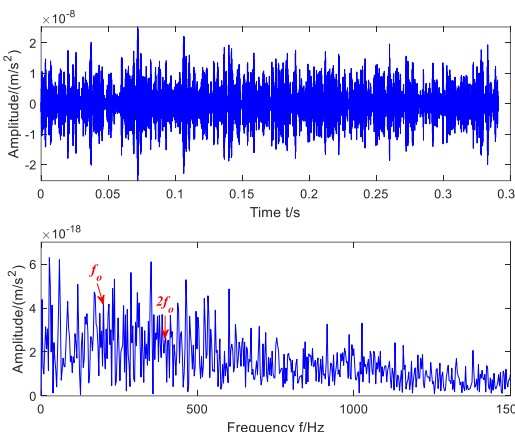

(**c**) Optimal component (above) and envelope spectrum (below) by Kurt index.

**Figure 7.** Comparison of optimal components obtained by three indexes.

### 4.3. Performance Analysis of SO-TVFEMD-EFIER-SVD

In order to analyze the feature extraction effect of the proposed SO-TVFEMD-ENEFI-SVD for bearing faults, a simulated fault signal of the inner ring is generated according to Equation (24). The fault signal of the inner ring is added with −1 dB Gaussian white noise, and the signal after adding noise and its envelope spectrum are shown in Figure 8.

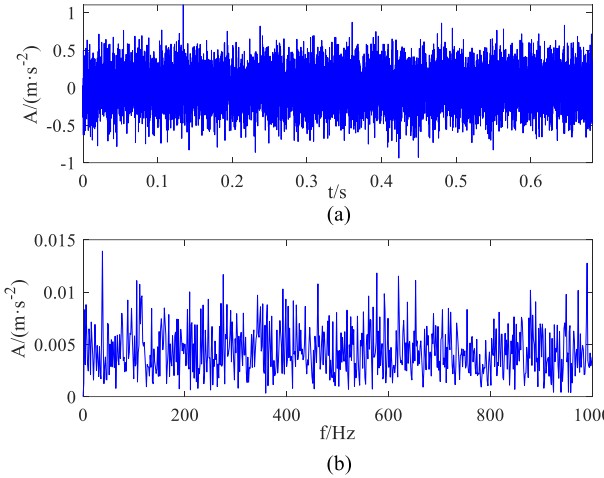

**Figure 8.** Simulation signal with noise: (**a**) waveform; (**b**) envelope spectrum.

Figure 6 shows the impact component, and the fault feature is completely engulfed by noise. The bearing fault type cannot be determined by the envelope spectrum in Figure 8. The simulation fault signal is analyzed by the proposed method. Firstly, the SO method is used to optimize the TVF-EMD's bandwidth threshold and B-spline order. The optimal EFIER (%) value of 2.9793 and the optimal parameters $(0.1507, 7)$ are found. Secondly, applying the optimal parameters TVF-EMD to decomposing the fault signal leads to 25 IMF components being produced. Figure 9 displays the results of calculating the EFIER values of each of the 25 IMF components. Then, the IMF components are fused according to the principle of the maximum EFIER. It can be seen from Figure 9 that we should first fuse IMF3 and IMF2, and then continue to fuse IMF10 until the optimal fusion component OC with the largest EFIER value is obtained. Table 4 displays the fusion results.

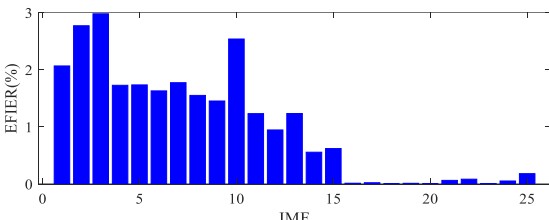

**Figure 9.** EFIER value of each IMF.

**Table 4.** IMF component integration of simulated bearing fault.

| Components | IMF3 | IMF3 + IMF2 | IMF3 + IMF2 + IMF10 |
|---|---|---|---|
| EFIER (%) | 2.9793 | 3.9959 | 3.8425 |

The following can be seen from Table 4:

(1) The EFIER value corresponding to IMF3 is the largest, denoted by MAX = 2.9793.

(2) The IMF3 and IMF2 are fused, and the EFIER value of the fused component is 3.9959, which is greater than MAX, and then MAX = 3.9959 is updated.

(3) Continue to fuse IMF10; if the fused component FEIER value is 3.8425, less than MAX, stop fusion

Therefore, the optimal component OC = IMF3 + IMF2. Figure 10 shows the IMF3 component's wave form and envelope spectrum. Figure 11 displays the waveform and envelope spectrum of the OC component (IMF3 + IMF2). In Figure 11, the fault characteristic frequencies $f_i$ and $2f_i$ can be observed, which can preliminarily determine the existence of

inner ring fault characteristics of the bearing. Figure 12 makes it easy to identify the fault characteristic frequencies $f_i$, $2f_i$, $3f_i$, $4f_i$, and $6f_i$. This suggests that the component fusion method suggested in this paper is capable of successfully fusing the fault information and preventing the loss of fault characteristic information.

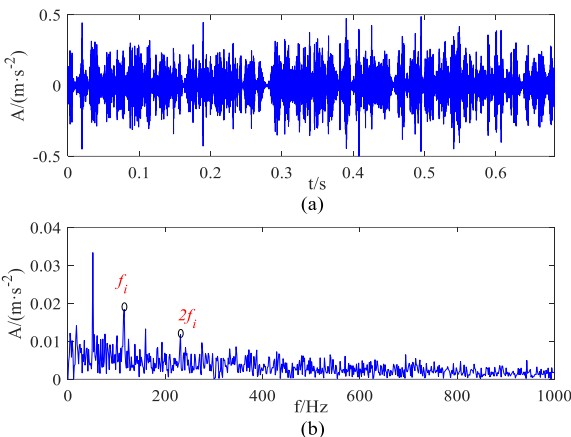

**Figure 10.** IMF3 component: (**a**) waveform; (**b**) envelope spectrum.

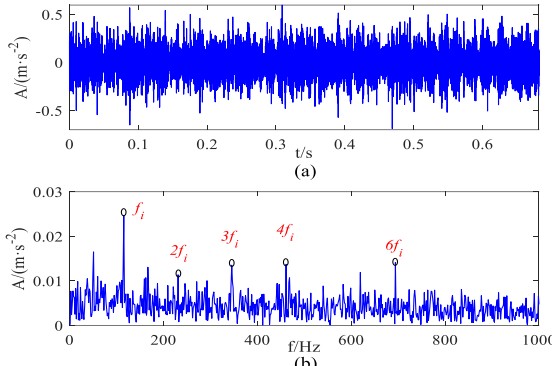

**Figure 11.** OC component (IMF3 + IMF2): (**a**) waveform; (**b**) envelope spectrum.

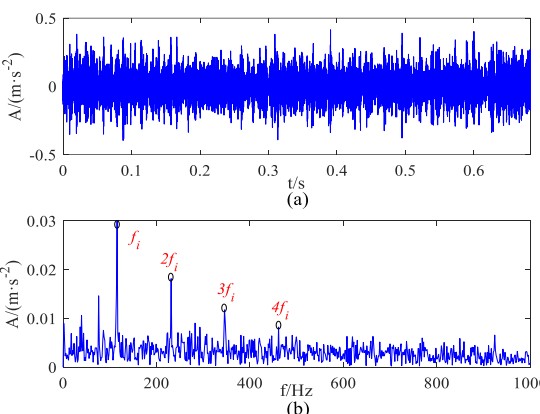

**Figure 12.** OC component after SVD denoising: (**a**) waveform; (**b**) envelope spectrum.

Finally, the OC component is denoised by SVD. Following noise reduction, Figure 12 displays the signal's waveform and envelope spectrum. Figure 12 clearly illustrates the markedly reduced noise in the OC component and the discernible fault characteristic frequencies of $f_i$, $2f_i$, $3f_i$, and $4f_i$. The inner ring fault impact features become more noticeable as the noise component is further weakened. The above simulation experiments

fully demonstrate that the early fault feature information can be successfully obtained using the suggested method.

## 5. Actual Bearing Fault Experiment

### 5.1. CWRU Dataset Expriment

The outer ring fault and inner ring fault signals of the bearing which are provided by CWRU [22] in the USA are used for experimental verification to confirm the efficacy of the proposed method. As illustrated in Figure 13, the experimental platform comprises a 1.5 KW motor, a torque sensor, a power tester, and an electronic controller. The specific parameters of the bearing are illustrated in https://engineering.case.edu/bearingdatacenter/apparatus-and-procedures, accessed on 8 June 2022. The experimental data in this research comprise the inner ring fault data from the drive end bearing fault data with a fault diameter of 0.1778 mm ($f_s = 48$ KHz, $f_o = 81.12$ Hz, and $f_i = 165.6$ Hz) (Table 5).

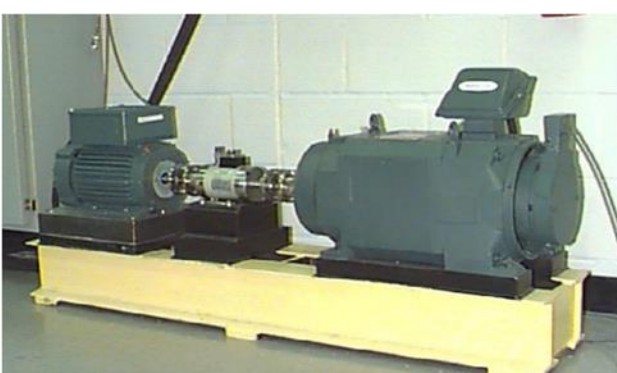

**Figure 13.** Experimental platforms of CWRU dataset.

**Table 5.** Bearing parameters.

| Parameters | Value |
| --- | --- |
| Ball diameter | 7.94 mm |
| Nodal circle diameter | 39.04 mm |
| Number of balls | 9 |
| Contact angle | 0° |

#### 5.1.1. Outer Ring Fault Diagnosis of CWRU Dataset

Figure 14 displays the waveform and envelope spectrum of the original outer ring fault vibration signal. From Figure 14, it can been see that the fault characteristics are fully submerged, making it challenging to obtain information on them. The periodic impact associated with the fault frequency cannot be observed, making it impossible to identify the bearing fault type.

The original vibration signal is analyzed using the proposed diagnosis method. First, the TVF-EMD optimization method's best parameter combination is determined by applying the SO algorithm, which yields a result of $(0.4819, 21)$. The seventh iteration yields the optimal combination of parameters, and 6.1258 is the highest EFIER value.

Second, TVF-EMD decomposes the original vibration signal into 12 IMF components. The EFIER values of the 12 IMF components are calculated, and Figure 14 shows the results. Then, the maximal envelope fault information energy ratio (EFIER) principle is utilized to fuse the IMF components. According to Figure 15, the EFIER value corresponding to IMF5 is the largest, so MAX = 6.1258. Then, the IMF5 and IMF4 components are fused, and the corresponding EFIER value is calculated. The EFIER value of the fusion component is 6.2036, which is greater than the MAX, so this is updated to MAX = 6.2036. The fusion

of the IMF1 component is continued; the EFIER value of the fusion component is 4.6459, which is less than the MAX, so the fusion is stopped. Therefore, the optimal component OC = IMF5 + IMF4. The relevant calculation process is shown in Table 6.

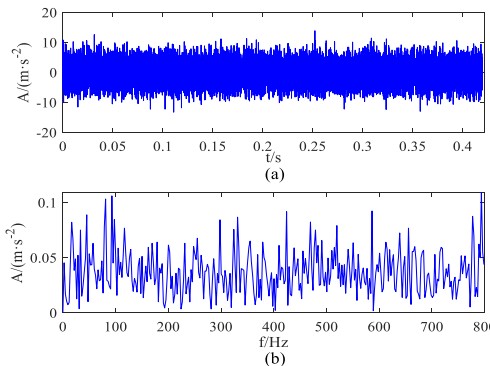

**Figure 14.** Original vibration signal of outer ring fault of CWRU dataset: (**a**) waveform; (**b**) envelope spectrum.

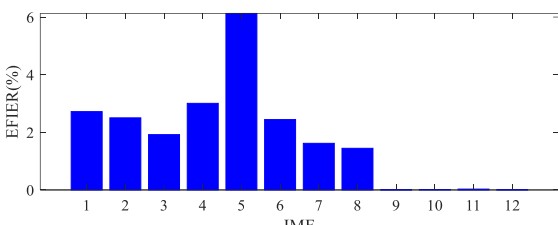

**Figure 15.** EFIER values of all IMF components.

**Table 6.** IMF component integration of outer ring fault of CWRU dataset.

| Components | IMF5 | IMF5 + IMF4 | IMF5 + IMF4 + IMF1 |
|:---:|:---:|:---:|:---:|
| EFIER (%) | 6.1258 | 6.2036 | 4.6459 |

Figure 16 illustrates the IMF5 component's waveform and envelope spectrum. The waveform and envelope spectrum of the optimal component OC are displayed in Figure 17. Figure 17 displays the fault characteristic frequencies $f_o$, $2f_o$, and $3f_o$. In Figure 17, the fault characteristic frequencies $f_o$, $2f_o$, $3f_o$, $4f_o$, and $5f_o$ are clearly visible. The OC component fused using the principle of the maximum EFIER contains more fault feature information, and the IMF5 component observes fewer fault characteristic frequencies.

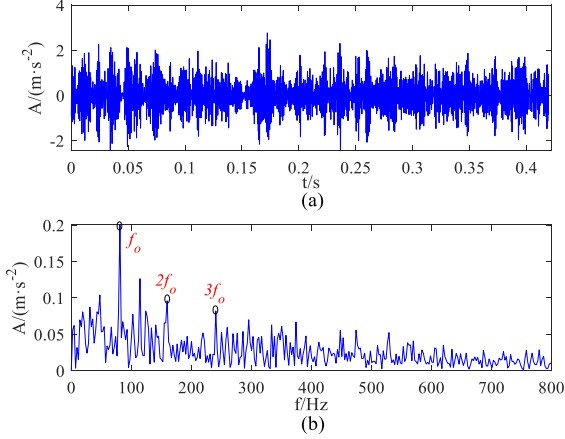

**Figure 16.** The IMF5 component: (**a**) waveform; (**b**) envelope spectrum.

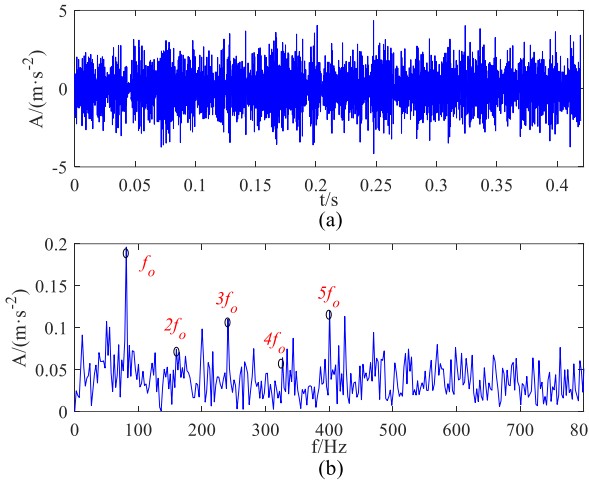

**Figure 17.** The OC component: (**a**) waveform; (**b**) envelope spectrum.

Finally, the OC component is denoised using SVD. Figure 18 demonstrates the OC component's waveform and envelope spectrum after SVD denoising. The fault characteristic frequencies $f_o$, $2f_o$, $3f_o$, $5f_o$, $6f_o$, $7f_o$, $8f_o$, and $9f_o$ are plainly visible in the noise reduction signal's envelope spectrum, and the fault characteristic information is very prominent. Therefore, the case study effectively demonstrates the viability of the modal component fusion method and the superiority of the improved TVF-EMD and SVD method in fault feature extraction.

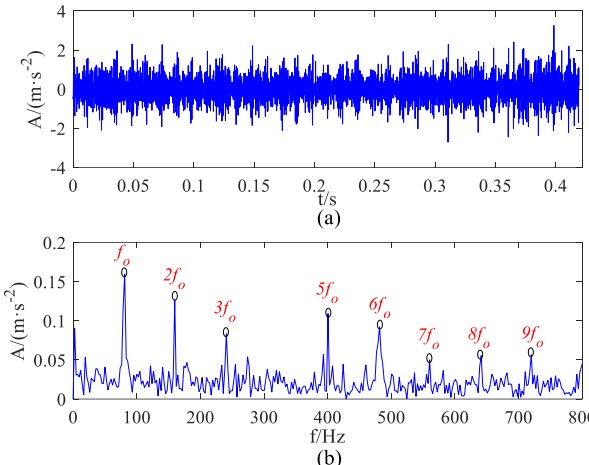

**Figure 18.** OC after SVD denoising: (**a**) waveform; (**b**) envelope spectrum.

Next, the traditional TVF-EMD, VMD, and CEEMDAN are used to experimentally evaluate the same signals. The default settings $(0.1, 26)$ are utilized for both parameters in the traditional TVF-EMD method. With the IMF11 component, EFIER = 4.4926, which is the largest EFIER value produced by the decomposition. The envelope spectrum and waveform of IMF11 are displayed in Figure 19. Figure 19 shows the fault characteristic frequency $f_o$.

In the VMD method, default values are used for the two parameters ($K = 3$ and $\alpha = 2000$). The VMD decomposes the vibration signal into three modes, and the kurtosis values are 3.0880, 3.0174, and 2.9985, respectively, with $u_1$ being the optimal component. Figure 20 shows the optimal modes obtained using the traditional VMD decomposition and the corresponding envelope spectrum. Figure 20 only shows the fault characteristic frequencies $f_o$ and $3f_o$, and the extracted fault feature information is not rich.

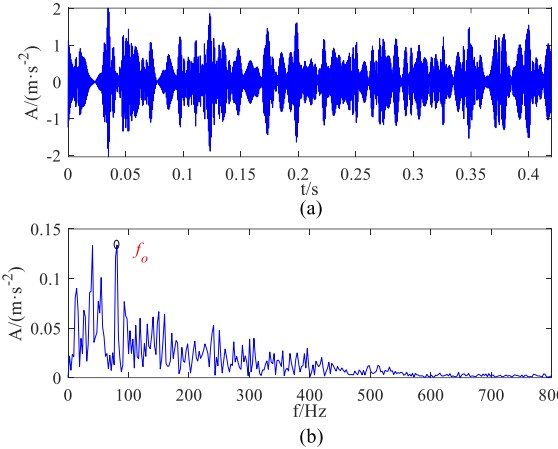

**Figure 19.** Optimal component by TVF-EMD: (**a**) waveform; (**b**) envelope spectrum.

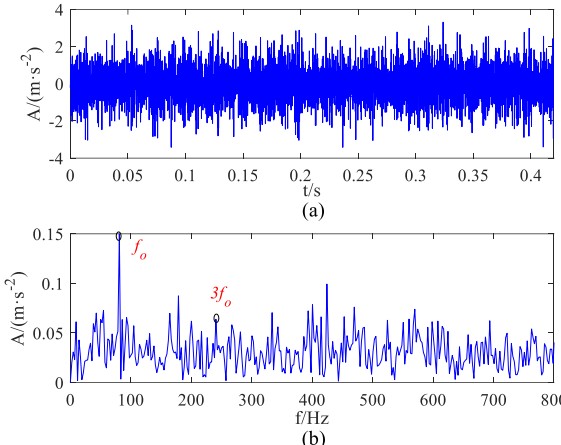

**Figure 20.** OC component by VMD decomposition: (**a**) waveform; (**b**) envelope spectrum.

The CEEMDAN method is used to decompose the signal. Figure 21 shows the waveform and envelope spectrum of the optimal component. In Figure 21, $f_o$ and $6f_o$ can be observed, and the extraction effect is not obvious. When using the improved TVF-EMD and SVD method, the extracted fault features are richer, and the fault diagnostic outcomes are more convincing. In contrast, the fault feature frequency extracted using the other methods is seriously affected by other irrelevant information, resulting in inaccurate diagnosis results.

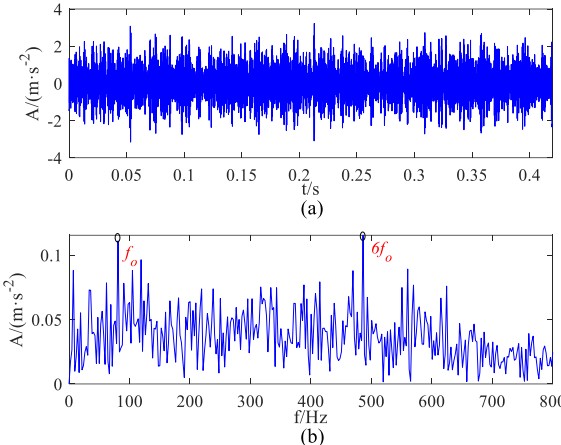

**Figure 21.** The optimal component by CEEMDAN: (**a**) waveform; (**b**) envelope spectrum.

Additionally, the EFIER values are used for a comparison in order to impartially assess these approaches' capacity to extract early fault features, and the findings are displayed in Table 7. Compared with the alternative methods, the EFIER value of the optimal component extracted using the method proposed in this paper is the largest, being 38.08%, 28.79%, and 73.51% higher than that of the other three methods. This shows that the improved TVF-EMD and SVD method can extract rich early fault feature information.

**Table 7.** Outer ring fault EFIER values of the four methods.

| Methods | Proposed Method | Traditional TVF-EMD | VMD | CEEMDAN |
|---------|-----------------|---------------------|-----|---------|
| EFIER (%) | 6.2036 | 4.4926 | 4.8167 | 3.5754 |

5.1.2. Inner Ring Fault Diagnosis

The waveform and envelope spectrum of the selected inner ring fault signal of CWRU are shown as Figure 22, and the inner fault frequency is $f_i = 165.6$ Hz. As shown in Figure 22, noise interference completely buries the fault impact information, making it impossible to identify the type of bearing failure or even locate the inner ring fault characteristic frequency in the envelope spectrum of the original vibration signal.

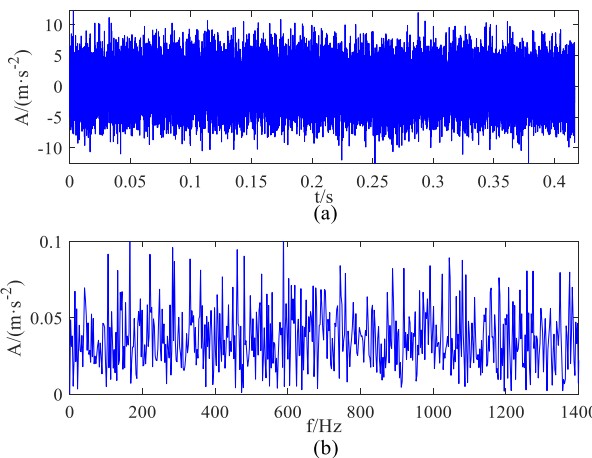

**Figure 22.** Original vibration signal of inner ring fault of CWRU dataset: (**a**) waveform; (**b**) envelope spectrum.

The original vibration signal is analyzed using the proposed diagnosis method. Firstly, during the ninth iteration, the optimal EFIER (%) value is 3.8966, and the optimal parameter combination is determined to be $(0.5444, 17)$. Secondly, TVF-EMD decomposes the original vibration signal into 13 IMF components. The EFIER values of the 13 IMF components are calculated, and Figure 23 shows the results.

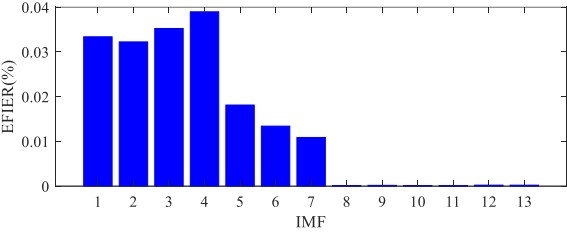

**Figure 23.** Every IMF component's EFIER value.

Then, the IMF components are fused. In Figure 23, it can be seen that the EFIER value corresponding to IMF4 is the largest, so MAX = 3.8966. Then, IMF4 and IMF3 are fused. The

fused component's EFIER value is 4.0580, which is greater than the MAX, so this is updated to MAX = 4.0580. The fusion of IMF1 is continued; the fused component's EFIER value is 3.6013, less than the MAX, so the fusion is stopped. Therefore, the optimal component OC = IMF4 + IMF3. The relevant calculation process is shown in Table 8.

**Table 8.** IMF component integration of inner ring fault of CWRU dataset.

| Components | IMF4 | IMF4 + IMF3 | IMF4 + IMF3 + IMF1 |
|---|---|---|---|
| EFIER (%) | 3.8966 | 4.0580 | 3.6013 |

Figure 24 illustrates the IMF5 component's waveform and envelope spectrum. The waveform and envelope spectrum of the optimal component OC are displayed in Figure 25. Only the fault characteristic frequencies $f_i$, $2f_i$, and $4f_i$ can be roughly identified in Figure 24, and the fault characteristics are not immediately apparent. In Figure 25, $f_i$, $2f_i$, $3f_i$, $4f_i$, $5f_i$, $7f_i$, and $8f_i$ can be observed. Compared with Figure 24, Figure 25 shows the extraction of rich fault feature information, allowing for it to be determined that the bearing has an inner ring fault.

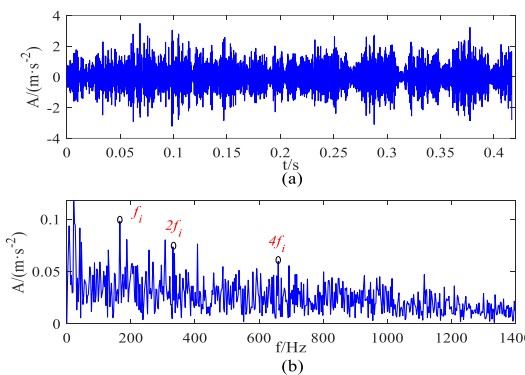

**Figure 24.** The IMF4 component: (**a**) waveform; (**b**) envelope spectrum.

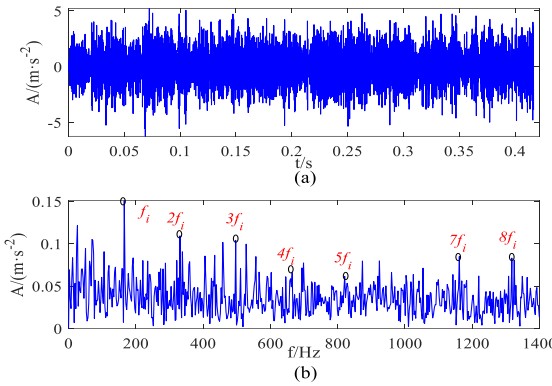

**Figure 25.** Optimal component obtained by ATVFEMD: (**a**) waveform; (**b**) envelope spectrum.

Finally, the OC component is denoised using SVD. Figure 26 demonstrates the OC component's waveform and envelope spectrum after SVD denoising. The fault characteristic frequencies $f_i$, $2f_i$, $3f_i$, $4f_i$, $5f_i$, $6f_i$, and $7f_i$ are easily visible. Therefore, the case study effectively demonstrates both the viability of the modal component fusion method and the superiority of the improved TVF-EMD and SVD method in fault feature extraction.

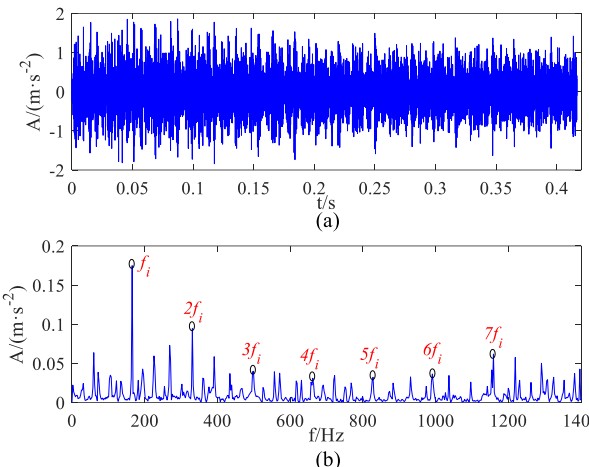

**Figure 26.** OC obtained using SO-TVFEMD and SVD denoising: (**a**) waveform; (**b**) envelope spectrum.

The parameters refer to case analysis A. In the traditional TVF-EMD method, with the IMF9 component, EFIER = 3.7704, which is the largest EFIER value produced by the decomposition. The envelope spectrum and waveform of IMF9 are displayed in Figure 27. Figure 27 shows the fault characteristic frequencies $f_i$ and $2f_i$.

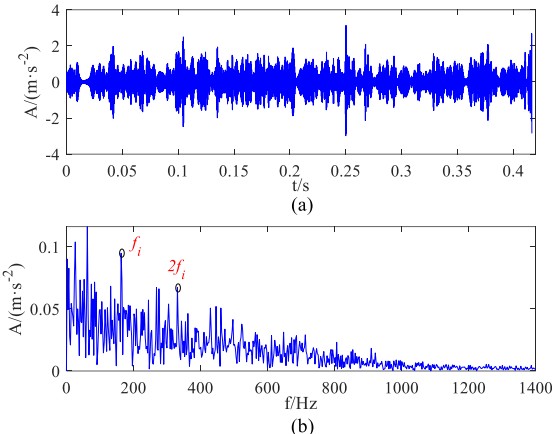

**Figure 27.** OC obtained by traditional TVF-EMD: (**a**) waveform; (**b**) envelope spectrum.

In VMD, the three decomposed modes have corresponding kurtosis values of 3.1032, 3.0767, and 2.9152, so $u_1$ is the optimal component. The optimal modes and matching envelope spectrum produced by VMD are displayed in Figure 28, and the obvious fault characteristic frequencies $f_i$ and $2f_i$ can be observed.

Figure 29 shows the waveform and envelope spectrum of the optimal component. Other fault characteristic information is not displayed in Figure 29, and only the fault characteristic frequency $f_i$ is visible.

Therefore, we can conclude that the fault feature information extracted using the other methods is relatively limited, and the improved TVF-EMD and SVD method can extract more clear and richer fault feature information than the other methods. In addition, we also use EFIER values to compare and quantitatively evaluate the ability of these methods to extract early fault features, the EFIER values of the optimal components processed by the four methods are calculated, respectively, and the results are shown in Table 9. Compared to alternative methods, the EFIER value of the optimal component extracted by the proposed method is the largest. Compared with alternative methods, the EFIER value of the optimal component extracted using the method proposed in this paper is the largest, being 7.63%,

15.85%, and 179.03% higher than that of the other three methods. This shows that the improved TVF-EMD and SVD method can extract rich early fault feature information.

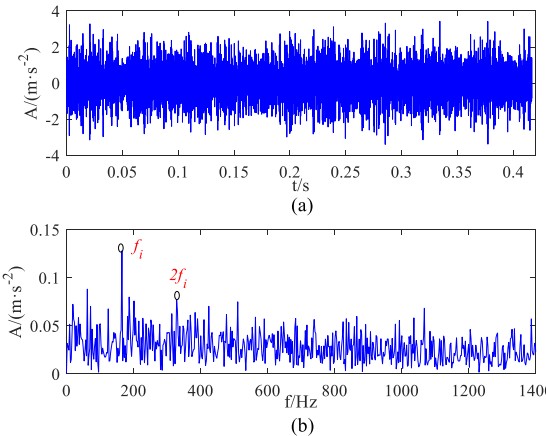

**Figure 28.** OC obtained by VMD: (**a**) waveform; (**b**) envelope spectrum.

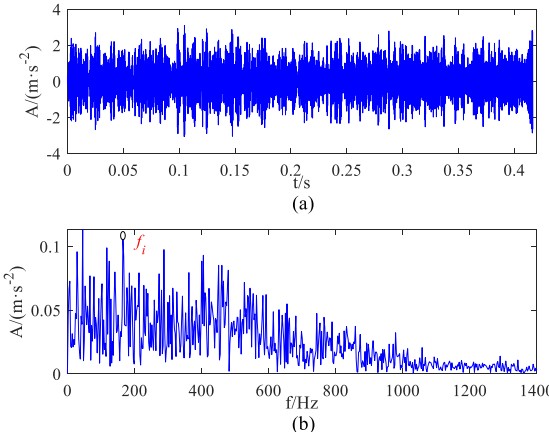

**Figure 29.** OC obtained using CEEMDAN: (**a**) waveform; (**b**) envelope spectrum.

**Table 9.** Inner ring fault EFIER values of the four methods.

| Methods | Proposed Method | Traditional TVF-EMD | VMD | CEEMDAN |
|---|---|---|---|---|
| EFIER (%) | 4.0580 | 3.7704 | 3.5028 | 1.4543 |

*5.2. XJTU Dataset Experiment*

In order to further verify the effectiveness of the method used in this paper, experimental analysis was conducted using rolling bearing fault data from XJTU of Xi'an Jiaotong University (https://cstr.cn/16666.11.nbsdc.bz283eWP, accessed on 16 March 2023). The bearing model used in the test bench is the LDK UER204 rolling bearing, and its parameters can be found in reference [26].

This dataset experiment has three operating conditions, each of which is divided into five bearings with a sampling frequency of 48,828 Hz. Select the 453rd data of the Bearing2_1 bearing under operating condition 2 has been selected as the experimental signal for inner ring fault. At this time, the speed is 2250 r/min, the radial force is 11 KN, and the frequency of inner ring fault is $f_i = 118.88$ Hz. The 70th data of the Bearing1_1 bearing under operating condition 1 has been selected as the experimental signal for the outer ring fault. At this time, the rotational speed is 2100 r/min, the radial force is 12 KN, and the outer ring fault frequency is $f_o = 81.12$ Hz.

### 5.2.1. Outer Ring Fault Experiment of XJTU Dataset

Figure 30 shows the time-domain waveform and envelope spectrum of the original outer ring fault signal. From Figure 30, it can be seen that the fault characteristics are completely submerged, and no information related to the frequency of the fault characteristics can be observed. Thus, the type of bearing fault is determined to be impossible.

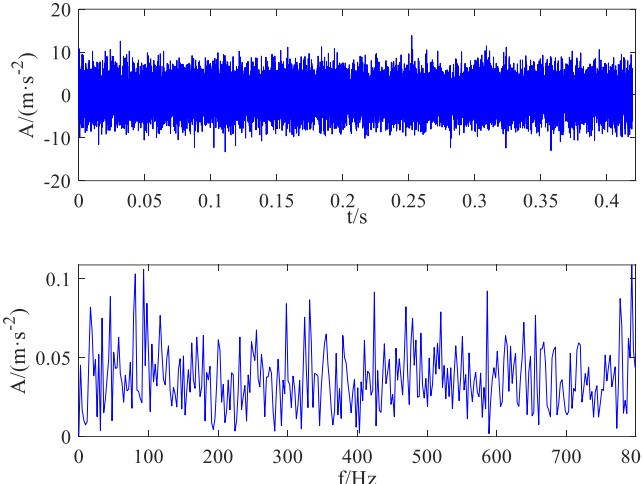

**Figure 30.** Waveform and envelope spectrum of outer ring fault of XJTU dataset.

Firstly, the SO algorithm is used to optimize TVF-EMD and obtain the optimal parameter combination $(\xi, n)$ of (0.4819,21). In the seventh iteration, the optimal parameter combination was obtained, and the maximum EFIER value corresponding to IMF5 is 6.1258%. Then, the original vibration signal was subjected to TVF-EMD decomposition using the optimal parameter combination (0.4819,21), resulting in 12 IMF. The EFIER values of 12 IMF components are separately calculated, as shown in Figure 31. Then, the IMF components are fused according to Figure 31. The relevant fusion process is shown in Table 10, and the obtained optimal fusion component is OC = IMF5 + IMF4.

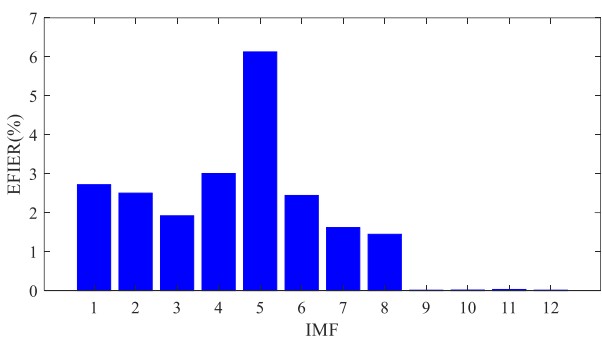

**Figure 31.** EFIER values of each IMF of outer ring fault of XJTU dataset.

**Table 10.** IMF component fusion process of IMF component.

| Component | IMF5 | IMF5 + IMF4 | IMF5 + IMF4 + IMF1 |
|---|---|---|---|
| EFIER (%) | 6.1258 | 6.2036 | 4.6459 |

Figure 32 shows the time-domain waveform and envelope spectrum of the optimal fusion component OC (IMF5 + IMF4). In Figure 32, the outer ring fault characteristic frequencies $f_o \sim 5f_o$ can be observed. It indicates that the fused component OC according to the EFIER maximum principle contains more fault characteristic information.

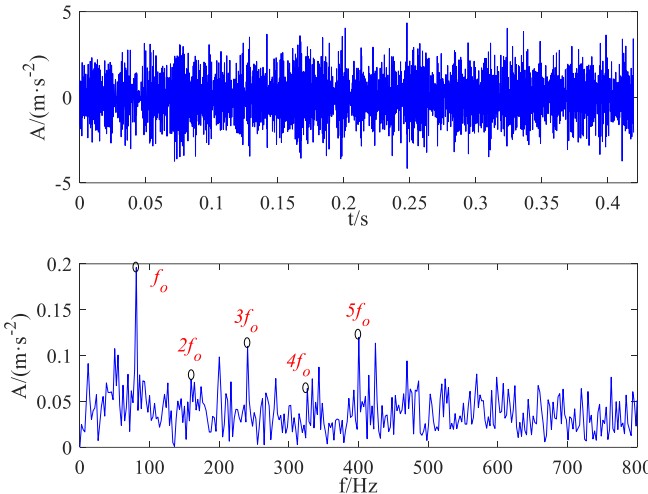

**Figure 32.** Waveform and envelope spectrum of OC after SVD denoising.

Then, the OC component can be denoised using SVD. Figure 33 shows the time-domain waveform and envelope spectrum of OC component after SVD denoise. It can be clearly observed from Figure 33 that the denoised envelope spectrum contains fault characteristic frequencies of the outer ring: $f_o \sim 3f_o$, $5f_o \sim 9f_o$.

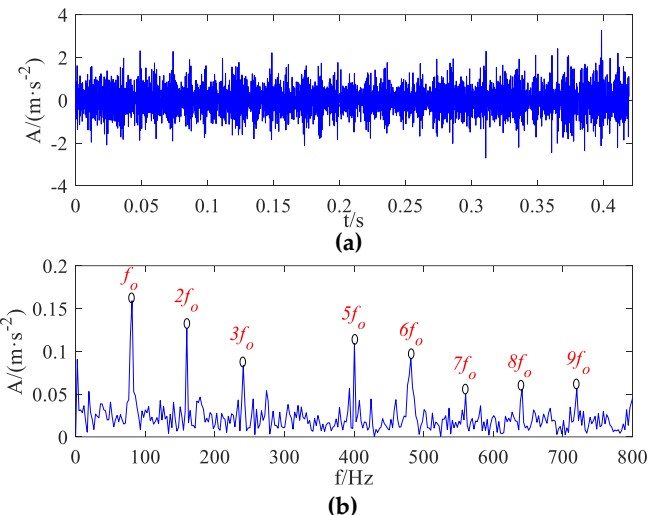

**Figure 33.** Denoised OC of XJTU outer ring fault by SVD: (**a**) waveform; (**b**) envelope spectrum.

The traditional TVF-EMD ($\xi = 0.1$, $n = 26$) method, VMD ($\alpha = 2000$, $K = 3$) method, and CEEMDAN method are used to extract early fault features of bearings, and the component with the highest EFIER value is selected as the sensitive component.

Figures 34–36 show the time-domain waveforms and envelope spectra of the sensitive components decomposed by the TVF-EMD, VMD, and CEEMDAN methods, respectively. From Figures 33–35, it can be seen that the fault feature information extracted using the method proposed in this paper (Figures 35 and 36) is more abundant with fewer noise interference components, and early fault features of bearings can be extracted more accurately by achieving the accurate identification of fault types. However, TVF-EMD, VMD, and CEEMDAN methods cannot effectively detect the characteristic frequencies of outer ring faults. The best performing VMD method can only observe $f_o$ and $5f_o$ and is greatly affected by noise, while other frequency multiplication information cannot be found.

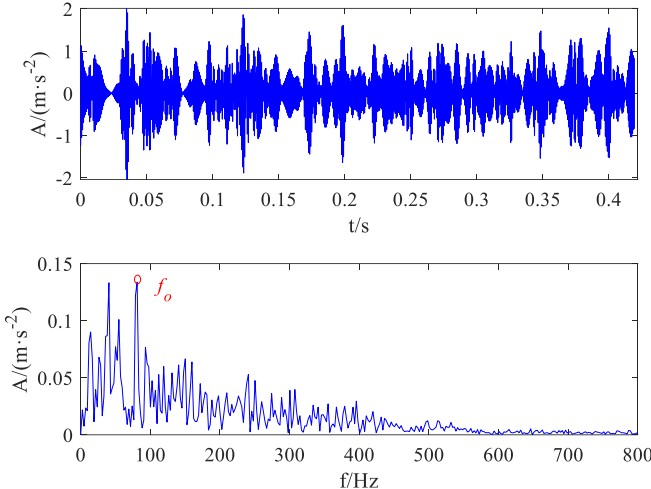

**Figure 34.** Waveform and envelope spectrum of optimal component by traditional TVF-EMD.

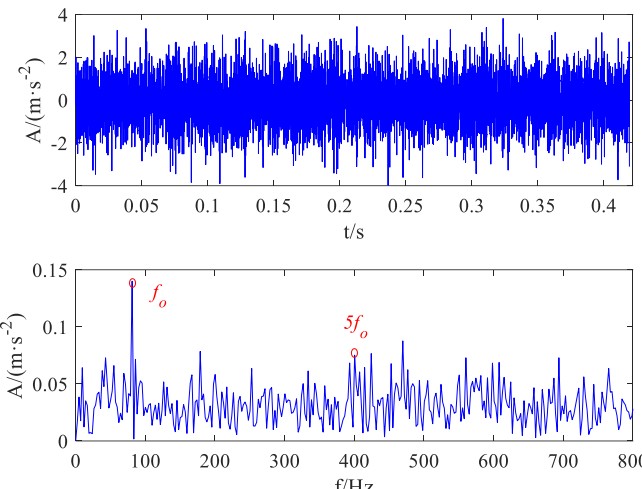

**Figure 35.** Waveform and envelope spectrum of OC of XJTU outer ring fault by VMD.

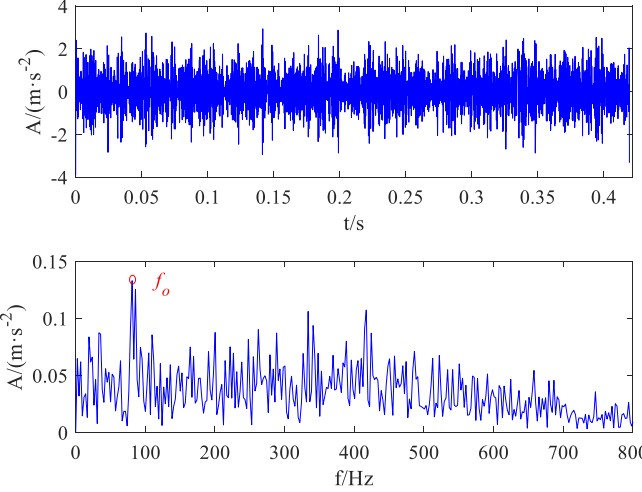

**Figure 36.** Waveform and envelope spectrum of optimal component by CEEMDAN.

### 5.2.2. Inner Ring Fault Experiment of XJTU Dataset

Figure 37 shows the time-domain waveform and envelope spectrum of the original inner ring fault signal. It can be seen that the complete submergence of fault characteristics makes it difficult to observe relevant fault information. The optimal parameter combination

$(\xi, n)$ is obtained as $(0,9)$ using the SO algorithm to optimize TVF-EMD. Subsequently, the original vibration signal was subjected to TVF-EMD decomposition using the optimal parameter combination, resulting in 50 IMF components. The maximum EFIER value corresponding to IMF43 is 2.8468. The EFIER values of the remaining IMF components are shown in Figure 38. The optimal fusion component OC obtained by fusing IMF components is IMF43 + IMF18, and the fusion process is shown in Table 11.

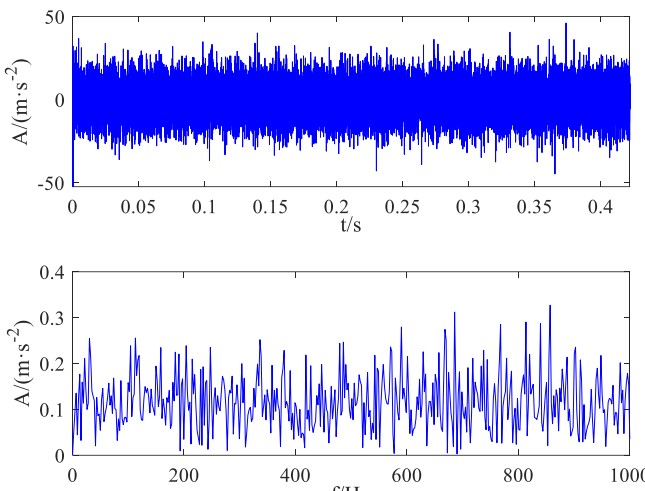

**Figure 37.** Waveform and envelope spectrum of inner ring fault of XJTU dataset.

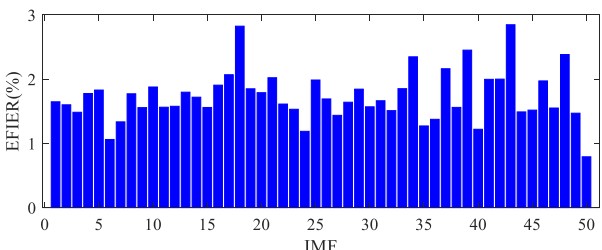

**Figure 38.** EFIER values of each IMF of inner ring fault of XJTU dataset.

**Table 11.** Fusion process of IMF components.

| Component | IMF43 | IMF43 + IMF18 | IMF43 + IMF18 + IMF39 |
|---|---|---|---|
| EFIER (%) | 2.8468 | 3.3613 | 2.5258 |

Figure 39 shows the time-domain waveform and envelope spectrum of the optimal fusion OC component (IMF43 + IMF18). In Figure 39, it can be observed that the fault features of the inner ring are $f_i \sim 4f_i$. Finally, SVD was used to denoise the OC component, and its time domain waveform and envelope spectrum are shown in Figure 40. From Figure 40, it can be seen that the noise component is significantly weakened, and it can be clearly observed that envelope spectrum after denoising contains fault characteristic frequencies of the inner ring $f_i \sim 2f_i$, $4f_i \sim 6f_i$, and $8f_i$.

The traditional TVF-EMD ($\xi = 0.1, n = 26$), VMD ($\alpha = 2000, K = 3$), and CCEMDAN methods continue to be used to extract early fault features of bearings while selecting the component with the maximum value of the EFIER as the sensitive component.

Figures 41–43 show the time-domain waveforms and envelope spectrum of the sensitive components decomposed by the TVF-EMD, VMD, and CEEMDAN methods, respectively. Comparing the results obtained by the proposed method with Figures 41–43, it can be seen that the fault feature information extracted by the method in this paper is richer

with fewer noise interference components and that the method in this paper can more accurately extract early fault features of bearings. For fault signals of the inner ring, the TVF-EMD method has the best effect among the TVF-EMD, VMD, and CEEMDAN methods. However, only the inner ring characteristic frequencies $2f_i$ and $3f_i$ can be observed in the results extracted by the TVF-EMD method, and it is not possible to find other frequency multiplications effectively due to noise interference.

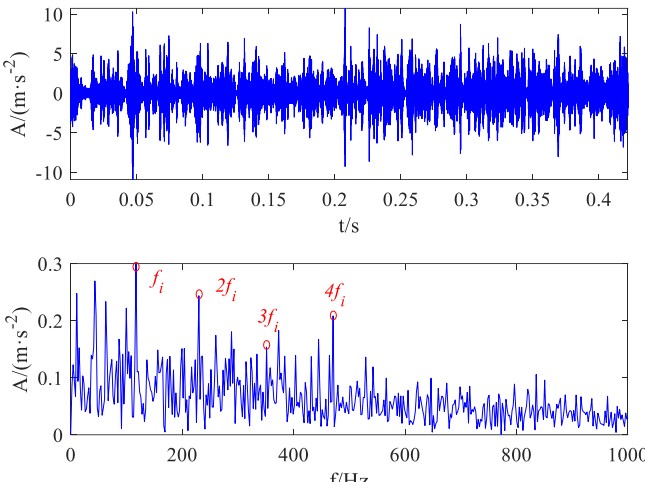

**Figure 39.** Waveform and envelope spectrum of OC component (IMF43 + IMF18).

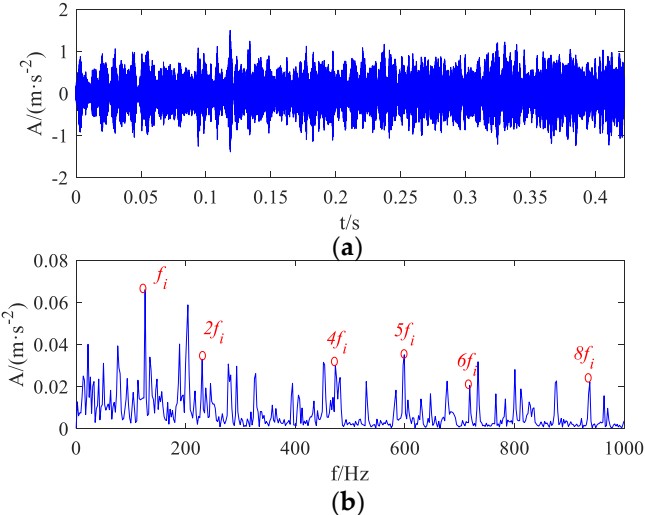

**Figure 40.** Denoised OC of XJTU inner ring fault; (**a**) waveform; (**b**) envelope spectrum.

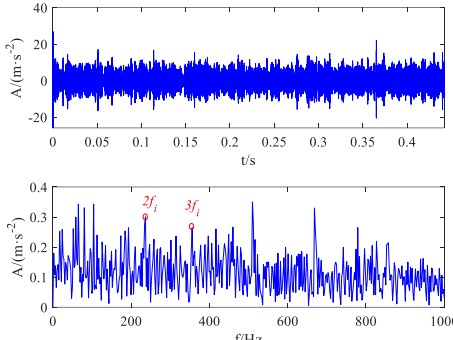

**Figure 41.** Waveform and envelope spectrum of OC by TVF-EMD.

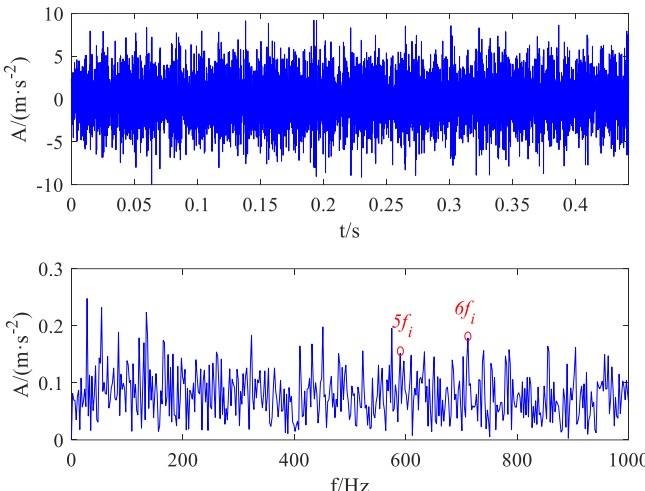

**Figure 42.** Waveform and envelope spectrum of OC of XJTU inner ring fault by VMD.

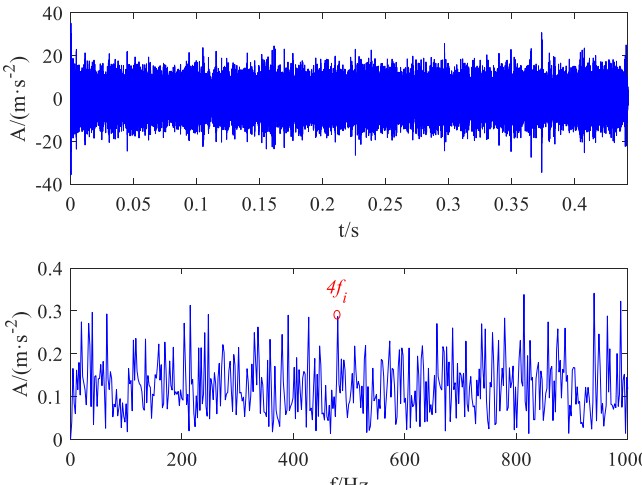

**Figure 43.** Waveform and envelope spectrum of OC by CEEMDAN.

The above experimental results further verify the effectiveness of our method in extracting early weak fault features of bearings. Compared with traditional TVF-EMD, VMD, and CEEMDAN methods, the results show that this method effectively improves the feature extraction ability of weak bearing faults. In the obtained envelope spectrum, the frequency multiplication peaks of the fault features are clear, and there are fewer interference frequencies near the peaks. This method has higher fault identification accuracy.

## 6. Conclusions

This study proposes a method for the early fault feature extraction of rolling bearings based on the adaptive fusion of TVF-EMD modal components and SVD noise reduction. The snake optimization (SO) technique is used to optimize the TVF-EMD algorithm in order to determine the optimal parameters that match the input signal. The primary conclusions of this study are as follows:

(1) The decomposition performance of the TVF-EMD algorithm and the settings of the two parameters (the bandwidth threshold and B-spline order) are crucial. Correctly deconstructing the original vibration signals is challenging if an irrational bandwidth threshold and B-spline order are applied, and significant under-decomposition, over-decomposition, or modal aliasing is highly likely to occur. Thus, the SO algorithm is used to optimize the TVF-EMD method.

(2) A new index, the envelope fault information energy ratio (EFIER), is proposed as the objective function. By comparing it with another index, it is proven that the EFIER is more robust to different noise intensities and burst pulses, in addition to having a greater sensitivity to fault feature information.

(3) The IMF components are fused using the principle of maximizing the energy ratio of the envelope fault information, and the optimal fusion components are obtained, thereby resolving the issue that the fault feature information following TVF-EMD decomposition is too dispersed and not concentrated enough.

(4) The optimal fusion components are noise-reduced using SVD in order to further reduce the impact of noise on the diagnostic results. A comparison is made between VMD, CEEMDAN, and the conventional TVF-EMD approaches. The methods can extract abundant information on early fault features, and the improved TVF-EMD and SVD methods outperform the other methods.

In the following work, we plan to further improve the proposed method from the following aspects:

(1) In the proposed method, the fault is identified manually mainly by the characteristic frequency of the fault signal after screening. In the next step, we hope to combine the work in this manuscript with deep learning to automatically identify bearing faults according to the characteristics of the filtered signals [27] and make a reasonable explanation of the identification process, so as to further improve the generalization and practicability of the work in this manuscript.

(2) At present, the proposed method has only carried out a lot of experimental analysis for a single fault, and there is not enough experimental verification for a compound fault. Therefore, in the next research, we will focus on applying the proposed method to the diagnosis and detection of compound faults with varying working conditions and varying loads and further verify the effectiveness of the proposed method on compound faults.

(3) When dealing with different types of fault signals, it is necessary to re-run the SO method for the parameter optimization of TVFEMD, resulting in a time-consuming optimization process. How to transplant the parameter optimization algorithm of TVFEMD between different datasets with the idea of the transfer learning algorithm, so as to improve the optimization efficiency, is also a problem that we will continue to study in depth.

**Author Contributions:** Conceptualization, X.E. and W.W.; methodology, X.E. and W.W.; validation, X.E. and H.Y.; data curation, H.Y. and W.W.; writing—original draft preparation, X.E.; writing—review and editing, X.E. and W.W. All authors have read and agreed to the published version of the manuscript.

**Funding:** This research was funded by the National Natural Science Foundation of China (No. 62173262), the "14th Five Year Plan" Hubei Provincial advantaged characteristic disciplines (groups) project of Wuhan University of Science and Technology (2023C0204), and Hubei Province Key Laboratory of Systems Science in Metallurgical Process (Wuhan University of Science and Technology) (Z202302).

**Data Availability Statement:** The data used are publicly available from CWRU and MFPT.

**Conflicts of Interest:** The authors declare no conflicts of interest.

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
