# Peer review of "Bearing Fault Feature Extraction Method Based on Adaptive Time-Varying Filtering Empirical Mode Decomposition and Singular Value Decomposition Denoising"

_machines, doi:10.3390/machines13010050_

Round 1
Reviewer 1 Report
Comments and Suggestions for Authors
The study represents a methodology composed of a mix of mathematical analysis tools to detect early failure in roller bearings.
The less convincing point in the research is that a set of standardised data was used to validate the results. The set of data has been extensively used by the research community and is a curated, well run, almost clinical, but fairly unrepresentative case for real world operations of machines using bearings. It should be a start point for research in detecting and extracting faults in bearings
The methodology is correct, and the implementation is sound. But it is just one of a myriad of combinations of tools and research papers from literature claiming that the results are marginally better.
I am not convinced the results would advance the knowledge and capability to detect and characterise faults in bearings.
The dataset is well known and needs to be properly attributed and referenced.
Author Response
Thanks very much for your suggestions. Please refer to document "Response-to-Reviewers-1-02.docx" for a detailed response

Reviewer 2 Report
Comments and Suggestions for Authors
The author used adaptive TVF-EDM and SVD denoising approach for the fault’s detection for the bearing fault. The paper is well written; however, the paper has some shortcomings, hence, I am recommending this paper for major review. The authors are requested to answer all the question point to point and highlight all the changes in the manuscript.
1. Please make a figure that describes the research methodology of this work, add this under the section of methodology.
2. I found some grammatical errors and punctuation errors. The title is also not adjusted well. Please correct them.
3. The authors have used various equations in this paper. What is the novelty of this work. If this is already published work, then what is the contribution of this work? Please don’t have too many mathematical equations, only put the most important.
4. What is the contribution of this work?
5. What are the future research opportunities in this area. Please mention it.
6. What is the significance of the proposed approach in terms of computational cost and applicability. How can this approach be applied for real-time fault detection, give an overview of that?
7. The authors are advised to add more relevant literature related fault detection. This is currently a developing field and various researchers around the world are working in this area. Comprehensive revision is needed in this regard. The introduction should also be revised to highlight the importance of machine learning, feature engineering, deep learning, and transfer learning, using the following as examples, Mechanical fault detection based on machine learning for robotic RV reducer using electrical current signature analysis; Prognostic health management of the robotic strain wave gear reducer based on variable speed of operation: a data-driven approach via deep learning; and Transfer learning-based intelligent fault detection approach for industrial robotic systems. For the deep learning model, you can refer to, Deep learning-based fault diagnosis of servo motor bearing using the attention-guided feature aggregation network.
8. Please mention the interpretable model for the future work in this paper you can refer to the recently published as, an explainable artificial intelligence-based approach for reliable damage detection in polymer composite structures using deep learning.
Author Response
Thanks very much for your suggestions. Please refer to document "Response-to-Reviewers-2-02.docx" for a detailed response

Reviewer 3 Report
Comments and Suggestions for Authors
- How robust is the SO optimization to variations in the initial conditions or different types of bearing faults? Does the optimal parameter set found for one fault type generalize well to other fault types, or is a separate optimization needed for each? Provide quantitative results demonstrating the robustness and generalization capability.
- How your proposed methodology is different from these papers? In these papers also, the authors have optimised the TVF-EMD using an optimization algorithm. Do consider these papers
· An amended grey wolf optimization with mutation strategy to diagnose bucket defects in Pelton wheel
· An ameliorated African vulture optimization algorithm to diagnose the rolling bearing defects
- The EFIER index is used for IMF component selection. What is the theoretical justification for this index, and how does its performance compare to other feature selection methods (e.g., mutual information, ReliefF)? Quantify the improvement in fault detection accuracy achieved by using EFIER versus a simpler selection criterion.
- While SVD is effective for noise reduction, it can also potentially attenuate weak fault features, especially in the early stages. How does the proposed method mitigate this risk?
- Provide a quantitative analysis demonstrating the trade-off between noise reduction and preservation of weak fault features for various signal-to-noise ratios (SNRs).
- The method is validated using two measured and simulation signals. However, how well does this method perform on a broader range of bearing types, operating conditions (e.g., different speeds, loads), and fault severities?
- What is the computational cost of the proposed method, and how scalable is it to larger datasets and more complex scenarios?
Author Response
Thanks very much for your suggestions. Please refer to document "Response-to-Reviewers-3-02.docx" for a detailed response

Reviewer 4 Report
Comments and Suggestions for Authors
The paper proposes an adaptive TVF-EMD model to enhance the clarity of fault indicators. It also proposes the EFIER index.
The value of the EFIER index for the proposed adaptive model scores highest as shown in Table 7. The numbers in Table 7 are not clear to which figures they refer to, are they figures 28, 29 and 30? Where is the figure related to the proposed method (the one you extract the EFIER of 4.0580)? Moreover, you have introduced a new adaptive model and a new index, is the EFIER index is only effective with the adaptive TVF-EMD model.
Is the adaptive TVF-EMD model effective and provides good results in terms of other indexes like kurtosis or crest factor of shape factor?
Line 368, what do you mean by the resistant term in the "EFIER is resistant"?
Line 468, what do you mean by simulative and simulation signals? is it the signal that is generated by a simulation model or is it the signals measured and generated from the experimental set? then, are sections 3 and 5 the same?
Where is section 4?
Why do you have Figure 21 in section 5.2 not in section 5.1? Is the experimental test used only for the inner fault?
In general, section 5 shall be reorganised in a way to make it easier for the reader. Compare the four methods for both cases: the inner and outer faults. Why don't you have like table 7 for the outer fault as well?
Line 712, you claim that "EFIER is more robust to weak noise", Does that mean it is not robust for high noise cases? Does that mean EFIER is effective in quiet applications and not industrial applications where noise is high?
I feel that the introduction of three pages is quite long.
Typo in line 120 Kurtosisis
Author Response
Thanks very much for your suggestions. Please refer to document "Response-to-Reviewers-4-02.docx" for a detailed response

Reviewer 5 Report
Comments and Suggestions for Authors
The authors have proposed a bearing fault feature extraction method based on adaptive TVF-EMD and SVD denoising in the present manuscript. The proposed method has used snake optimization for algorithm optimization. The manuscript is written well, and its overall contribution is clear. Here are a few suggestions for improving the manuscript further.
1) The author should elaborate on how dividing the bearing signal into several intrinsic mode functions (IMFs) by TVF-EMD reduces the nonlinearity and non-stationarity of the fault signal.
2) What is the motivation behind using the snake optimization? Have other optimization techniques like ant colony optimization or others been tested for the analysis?
3) The authors can study work based on the multi-size wide kernel convolutional neural network for bearing fault diagnosis and electric vehicle motor fault detection with improved recurrent 1d convolutional neural network.
4) More detailed comparative studies between VMD, CEEMDAN, and the conventional TVF-EMD approaches can be included.
5) What are the limitations of the proposed technique?
6) The legibility of all the figures should be improved.
Author Response
Thanks very much for your suggestions. Please refer to document "Response-to-Reviewers-5-02.docx" for a detailed response

Round 2
Reviewer 1 Report
Comments and Suggestions for Authors
I appreciate the authors' efforts to improve the manuscript.
I also appreciate the authors addressing the aspects I suggested were less than stellar in the previous version.
As a result of the work, I am satisfied the manuscript adds to the knowledge in the area bearing fault extraction and I recommend it is published in its present form.
Reviewer 2 Report
Comments and Suggestions for Authors
Thank you for highlighting all the points.
Comments on the Quality of English LanguageLooks fine
Reviewer 3 Report
Comments and Suggestions for Authors
No further comments